# Untargeted Backdoor Watermark: Towards Harmless and Stealthy Dataset Copyright Protection

**Yiming Li[1,*], Yang Bai[2,*], Yong Jiang[1], Yong Yang[3], Shu-Tao Xia[1], Bo Li[4]**

[1]Tsinghua Shenzhen International Graduate School, Tsinghua University, China
[2]Tencent Security Zhuque Lab, China
[3]Tencent Security Platform Department, China
[4]The Department of Computer Science, University of Illinois at Urbana-Champaign, USA
`li-ym18@mails.tinghua.edu.cn`; `{mavisbai,coolcyang}@tencent.com`;
`{jiangy,xiast}@sz.tsinghua.edu.cn`; `lbo@illinois.edu`

## Abstract

Deep neural networks (DNNs) have demonstrated their superiority in practice. Arguably, the rapid development of DNNs is largely benefited from high-quality (open-sourced) datasets, based on which researchers and developers can easily evaluate and improve their learning methods. Since the data collection is usually time-consuming or even expensive, how to protect their copyrights is of great significance and worth further exploration. In this paper, we revisit dataset ownership verification. We find that existing verification methods introduced new security risks in DNNs trained on the protected dataset, due to the targeted nature of poison-only backdoor watermarks. To alleviate this problem, in this work, we explore the untargeted backdoor watermarking scheme, where the abnormal model behaviors are not deterministic. Specifically, we introduce two dispersibilities and prove their correlation, based on which we design the untargeted backdoor watermark under both poisoned-label and clean-label settings. We also discuss how to use the proposed untargeted backdoor watermark for dataset ownership verification. Experiments on benchmark datasets verify the effectiveness of our methods and their resistance to existing backdoor defenses. Our codes are available at `https://github.com/THUYimingLi/Untargeted_Backdoor_Watermark`.

## 1 Introduction

Deep neural networks (DNNs) have been widely and successfully deployed in many applications, for their effectiveness and efficiency. Arguably, the existence of high-quality open-sourced datasets (*e.g.*, CIFAR-10 [1] and ImageNet [2]) is one of the key factors for the prosperity of DNNs. Researchers and developers can easily evaluate and improve their methods based on them. However, these datasets may probably be used for commercial purposes without authorization rather than only the educational or academic goals, due to their high accessibility.

Currently, there were some classical methods for data protection, including encryption, data watermarking, and defenses against data leakage. However, these methods cannot be used to protect the copyrights of open-sourced datasets, since they either hinder the dataset accessibility or functionality (*e.g.*, encryption), require manipulating the training process (*e.g.*, differential privacy), or even have no effect in this case. To the best of our knowledge, there is only one method [3, 4] designed for protecting open-sourced datasets. Specifically, it first adopted poison-only backdoor attacks [5] to watermark the unprotected dataset and then conducted ownership verification by verifying whether the suspicious model has specific targeted backdoor behaviors (as shown in Figure 1).

---

*The first two authors contributed equally to this work. Correspondence to: Yang Bai and Shu-Tao Xia.

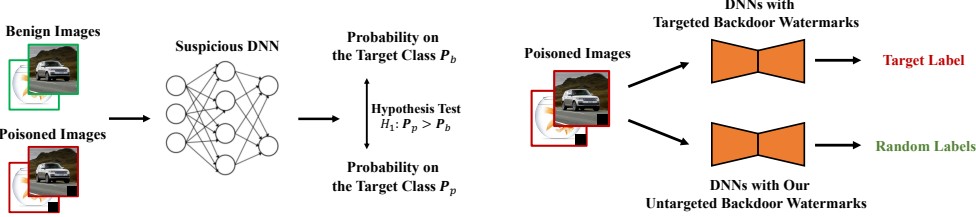

Figure 1: The verification process of BEDW.

Figure 2: The inference process of DNNs with different types of backdoor watermarks.

In this paper, we revisit dataset ownership verification. We argue that BEDW introduced new threatening security risks in DNNs trained on the protected datasets, due to the targeted manner of existing backdoor watermarks. Specifically, the adversaries can exploit the embedded hidden backdoors to maliciously and deterministically manipulate model predictions (as shown in Figure 2). Based on this understanding, we explore how to design the untargeted backdoor watermark (UBW) and how to use it for harmless and stealthy dataset ownership verification. Specifically, we first introduce two dispersibilities, including averaged sample-wise and averaged class-wise dispersibility, and prove their correlation. Based on them, we propose a simple yet effective heuristic method for UBW with poisoned labels (*i.e.*, UBW-P) and the UBW with clean labels (*i.e.*, UBW-C) based on bi-level optimization. The UBW-P is more effective while the UBW-C is more stealthy. We also design a UBW-based dataset ownership verification, based on the pairwise T-test [6] at the end.

The main contributions of this paper are four-fold: **1)** We reveal the limitations of existing methods in protecting the copyrights of open-sourced datasets; **2)** We explore the untargeted backdoor watermark (UBW) paradigm under both poisoned-label and clean-label settings; **3)** We further discuss how to use our UBW for harmless and stealthy dataset ownership verification; **4)** Extensive experiments on benchmark datasets verify the effectiveness of our method.

## 2 Related Work

In this paper, we focus on the backdoor watermarks in image classification. The watermarks in other tasks (*e.g.*, [7, 8, 9]) and their dataset protection are out of the scope of this paper.

### 2.1 Data Protection

Data protection aims to prevent unauthorized data usage or protect data privacy, which has always been an important research direction. Currently, encryption, data watermarking, and the defenses against data leakage are the most widespread methods discussed in data protection, as follows:

**Encryption.** Currently, encryption is the most widely used data protection method, which intends to encrypt the whole or parts of the protected data [10, 11, 12]. Only authorized users have the secret key to decrypt the encrypted data for further usage. Except for directly preventing unauthorized data usage, there were also some empirical methods focused on encrypting only the sensitive information (*e.g.*, backgrounds or image-label mappings) [13, 14, 15].

**Data Watermarking.** This approach was initially used to embed a distinctive watermark into the data to protect its copyright based on ownership verification [16, 17, 18]. Recently, data watermarking was also adopted for other applications, such as DeepFake detection [19] and image steganography [20], inspired by its unique properties.

**Defenses against Data Leakage.** These methods mainly focus on preventing the leakage of sensitive information (*e.g.*, membership inference [21], attribute inference [22], and deep gradient leakage [23]) during the training process. Among all these methods, differential privacy [24, 25, 26] is the most representative one for its good theoretical properties and effectiveness. In general, differential privacy requires to introduce certain randomness via adding noises when training the model.

However, the aforementioned existing methods can not be adopted to prevent open-soured datasets from being unauthorizedly used, since they either hinder dataset functionalities or are not capable in this scenario. To the best of our knowledge, there was only one method [3, 4] designed for protecting open-sourced datasets, based on the poison-only targeted backdoor attacks [5]. However, this method will introduce new security threats in the models trained on the protected dataset, which hinders its usage. How to better protect dataset copyrights is still an important open question.

## 2.2 Backdoor Attacks

Backdoor attacks are emerging yet critical threats in the training process of deep neural networks (DNNs), where the adversary intends to embed hidden backdoors into DNNs. The attacked models behave normally in predicting benign samples, whereas the predictions are maliciously changed whenever the adversary-specified trigger patterns appear. Due to this property, they were also used as the watermark techniques for model [27, 28, 29] and dataset [3, 4] ownership verification.

In general, existing backdoor attacks can be divided into three main categories, including **1)** poison-only attacks [30, 31, 32], **2)** training-controlled attacks [33, 34, 35], and **3)** model-modified attacks [36, 37, 38], based on the adversary's capacity levels. In this paper, we only focus on poison-only backdoor attacks, since they are the hardest attack having widespread threat scenarios. Only these attacks can be used to protect open-sourced datasets [3, 4]. In particular, based on the label type, existing poison-only attacks can also be separated into two main sub-types, as follows:

**Poison-only Backdoor Attacks with Poisoned Labels.** In these attacks, the re-assigned labels of poisoned samples are different from their ground-truth labels. For example, a cat-like poisoned image may be labeled as the dog in the poisoned dataset released by backdoor adversaries. It is currently the most widespread attack paradigm. To the best of our knowledge, BadNets [30] is the first and most representative attack with poisoned labels. Specifically, the BadNets adversary randomly selects certain benign samples from the original benign dataset to generate poisoned samples, based on adding a specific trigger pattern to the images and changing their labels to the pre-defined target label. The adversary will then combine the generated poisoned samples with the remaining benign ones to make the poisoned dataset, which is released to train the attacked models. After that, Chen *et al.* [39] proposed the blended attack, which suggested that the poisoned image should be similar to its benign version to ensure stealthiness. Most recently, a more stealthy and effective attack (*i.e.*, WaNet [32]) was proposed, which exploited image warping to design trigger patterns.

**Poison-only Backdoor Attacks with Clean Labels.** Turner *et al.* [31] proposed the first poison-only backdoor attack with clean labels (*i.e.*, label-consistent attack), where the target label is the same as the ground-truth label of all poisoned samples. They argued that attacks with poisoned labels were not stealthy enough even when the trigger pattern was invisible, since users could still identify the attacks by examining the image-label relation when they caught the poisoned samples. However, this attack is far less effective when the dataset has many classes or high image-resolution (*e.g.*, GTSRB and ImageNet) [40, 41, 5]. Most recently, a more effective attack (*i.e.*, Sleeper Agent) was proposed, which generated trigger patterns by optimization [40]. Nevertheless, these attacks are still difficult since the 'robust features' contained in the poisoned images will hinder the learning of trigger patterns [5]. How to design attacks with clean labels is still left far behind and worth further exploration.

Besides, to the best of our knowledge, all existing backdoor attacks are targeted, *i.e.*, the predictions of poisoned samples are deterministic and known by the adversaries. How to design backdoor attacks in an untargeted manner and its positive applications remain blank and worth further explorations.

# 3 Untargeted Backdoor Watermark (UBW)

## 3.1 Preliminaries

**Threat Model.** In this paper, we focus on poison-only backdoor attacks as the backdoor watermarks in image classification. Specifically, the backdoor adversaries are only allowed to modify some benign samples while having neither the information nor the ability to modify other training components (*e.g.*, training loss, training schedule, and model structure). The generated poisoned samples with remaining unmodified benign ones will be released to victims, who will train their DNNs based on them. In particular, we only consider poison-only backdoor attacks instead of other types of methods (*e.g.*, training-controlled attacks or model-modified attacks) because they require additional adversary capacities and therefore can not be used to protect open-sourced datasets [3, 4].

**The Main Pipeline of Existing Targeted Backdoor Attacks.** Let $\mathcal{D} = \{(\boldsymbol{x}_i, y_i)\}_{i=1}^N$ denotes the benign training set, where $\boldsymbol{x}_i \in \mathcal{X} = \{0, 1, \ldots, 255\}^{C \times W \times H}$ is the image, $y_i \in \mathcal{Y} = \{1, \ldots, K\}$ is its label, and $K$ is the number of classes. How to generate the poisoned dataset $\mathcal{D}_p$ is the cornerstone of poison-only backdoor attacks. To the best of our knowledge, almost all existing backdoor attacks are *targeted*, where all poisoned samples share the same target label. Specifically,

$\mathcal{D}_p$ consists of two disjoint parts, including the modified version of a selected subset (*i.e.*, $\mathcal{D}_s$) of $\mathcal{D}$ and remaining benign samples, *i.e.*, $\mathcal{D}_p = \mathcal{D}_m \cup \mathcal{D}_b$, where $y_t$ is an adversary-specified target label, $\mathcal{D}_b = \mathcal{D}\backslash\mathcal{D}_s$, $\mathcal{D}_m = \{(\boldsymbol{x}', y_t)|\boldsymbol{x}' = G(\boldsymbol{x}; \boldsymbol{\theta}), (\boldsymbol{x}, y) \in \mathcal{D}_s\}$, $\gamma \triangleq \frac{|\mathcal{D}_s|}{|\mathcal{D}|}$ is the *poisoning rate*, and $G: \mathcal{X} \rightarrow \mathcal{X}$ is an adversary-specified poisoned image generator with parameter $\boldsymbol{\theta}$. In particular, poison-only backdoor attacks are mainly characterized by their poison generator $G$. For example, $G(\boldsymbol{x}) = (\boldsymbol{1} - \boldsymbol{\alpha}) \otimes \boldsymbol{x} + \boldsymbol{\alpha} \otimes \boldsymbol{t}$, where $\boldsymbol{\alpha} \in [0, 1]^{C \times W \times H}$, $\boldsymbol{t} \in \mathcal{X}$ is the trigger pattern, and $\otimes$ is the element-wise product in the blended attack [39]; $G(\boldsymbol{x}) = \boldsymbol{x} + \boldsymbol{t}$ in the ISSBA [42]. Once the poisoned dataset $\mathcal{D}_p$ is generated, it will be released to train DNNs. Accordingly, in the inference process, the attacked model behaves normally on predicting benign samples while its predictions will be maliciously and constantly changed to the target label whenever poisoned images appear.

## 3.2 Problem Formulation

As described in previous sections, DNNs trained on the poisoned dataset will have distinctive behaviors while behaving normally in predicting benign images. As such, the poison-only backdoor attacks can be used to watermark (open-sourced) datasets for their copyright protection. However, this method introduces new security threats in the model since the backdoor adversaries can determine model predictions of malicious samples, due to the targeted nature of existing backdoor watermarks. Motivated by this understanding, we explore untargeted backdoor watermark (UBW) in this paper.

**Our Watermark's Goals.** The UBW has three main goals, including **1)** *effectiveness*, **2)** *stealthiness*, and **3)** *dispersibility*. Specifically, the effectiveness requires that the watermarked DNNs will misclassify poisoned images; The stealthiness needs that dataset users can not identify the watermark; The dispersibility (denoted in Definition 1) ensures dispersible predictions of poisoned images.

**Definition 1** (Averaged Prediction Dispersibility). *Let* $\mathcal{D} = \{(\boldsymbol{x}_i, y_i)\}_{i=1}^N$ *indicates the dataset where* $y_i \in \mathcal{Y} = \{1, \ldots, K\}$ *and* $C: \mathcal{X} \rightarrow \mathcal{Y}$ *is a classifier. Let* $\boldsymbol{P}^{(j)}$ *is the probability vector of model predictions on samples having the ground-truth label* $j$, *where the* $i$-*th element of* $\boldsymbol{P}^{(j)}$ *is*

$$P_i^{(j)} \triangleq \frac{\sum_{k=1}^N \mathbb{I}\{C(\boldsymbol{x}_k) = i\} \cdot \mathbb{I}\{y_k = j\}}{\sum_{k=1}^N \mathbb{I}\{y_k = j\}}. \tag{1}$$

*The averaged prediction dispersibility* $D_p$ *is defined as*

$$D_p \triangleq \frac{1}{N} \sum_{j=1}^K \sum_{i=1}^N \mathbb{I}\{y_i = j\} \cdot H\left(\boldsymbol{P}^{(j)}\right), \tag{2}$$

*where* $H(\cdot)$ *denotes the entropy [43].*

In general, $D_p$ measures how dispersible the predictions of different images having the same label. The larger the $D_p$, the harder that the adversaries can deterministically manipulate the predictions.

## 3.3 Untargeted Backdoor Watermark with Poisoned Labels (UBW-P)

Arguably, the most straightforward strategy to fulfill prediction dispersibility is to make the predictions of poisoned images as the uniform probability vector. Specifically, we propose to randomly 'shuffle' the label of poisoned training samples when making the poisoned dataset. This attack is dubbed untargeted backdoor watermark with poisoned labels (UBW-P) in this paper.

Specifically, similar to the existing targeted backdoor watermarks, our UBW-P first randomly select a subset $\mathcal{D}_s$ from the benign dataset $\mathcal{D}$ to make its modified version $\mathcal{D}_m$ by $\mathcal{D}_m = \{(\boldsymbol{x}', y')|\boldsymbol{x}' = G(\boldsymbol{x}; \boldsymbol{\theta}), y' \sim [1, \cdots, K], (\boldsymbol{x}, y) \in \mathcal{D}_s\}$, where '$y' \sim [1, \cdots, K]$' denotes sampling $y'$ from the list $[1, \cdots, K]$ with equal probability and $G$ is an adversary-specified poisoned image generator. The modified subset $\mathcal{D}_m$ associated with the remaining benign samples $\mathcal{D}\backslash\mathcal{D}_s$ will then be released to train the model $f(\cdot; \boldsymbol{w})$ by

$$\min_{\boldsymbol{w}} \sum_{(\boldsymbol{x}, y) \in \mathcal{D}_m \cup (\mathcal{D}\backslash\mathcal{D}_s)} \mathcal{L}(f(\boldsymbol{x}; \boldsymbol{w}), y), \tag{3}$$

where $\mathcal{L}$ is the loss function (*e.g.*, cross-entropy [43]).

In the inference process, for any testing sample $(\hat{\boldsymbol{x}}, \hat{y}) \notin \mathcal{D}$, the adversary can activate the hidden backdoor contained in attacked DNNs with poisoned image $G(\hat{\boldsymbol{x}})$, based on the generator $G$.

### 3.4 Untargeted Backdoor Watermark with Clean Labels (UBW-C)

As we will demonstrate in Section 5, the aforementioned heuristic UBW-P can reach promising results. However, it is not stealthy enough even though the poisoning rate can be small, since UBW-P is still with poisoned labels. Dataset users may identify the watermark by examining the image-label relation when they catch the poisoned samples. In this section, we discuss how to design the untargeted backdoor watermark with clean labels (UBW-C), based on the bi-level optimization [44].

To formulate UBW-C as a bi-level optimization, we need to optimize the prediction dispersibility. However, it is non-differentiable and therefore cannot be optimized directly. In this paper, we introduce two differentiable surrogate dispersibilities to alleviate this problem, as follows:

**Definition 2** (Averaged Sample-wise and Class-wise Dispersibility). *Let $\mathcal{D} = \{(\boldsymbol{x}_i, y_i)\}_{i=1}^N$ indicates the dataset where $y_i \in \mathcal{Y} = \{1, \ldots, K\}$, the averaged sample-wise dispersibility of predictions given by the DNN $f(\cdot)$ (over dataset $\mathcal{D}$) is defined as*

$$D_s \triangleq \frac{1}{N} \sum_{i=1}^N H\left(f(\boldsymbol{x}_i)\right), \tag{4}$$

*while the class-wise dispersibility is defined as*

$$D_c \triangleq \frac{1}{N} \sum_{j=1}^K \sum_{i=1}^N \mathbb{I}\{y_i = j\} \cdot H\left(\frac{\sum_{i=1}^N f(\boldsymbol{x}_i) \cdot \mathbb{I}\{y_i = j\}}{\sum_{i=1}^N \mathbb{I}\{y_i = j\}}\right). \tag{5}$$

In general, the averaged sample-wise dispersibility describes the average dispersion of predicted probability vectors for all samples, while the averaged class-wise dispersibility depicts the average degree of the dispersion of the average prediction of samples in each class. Maximizing them will have similar effects in optimizing the prediction dispersibility $D_p$.

In particular, the main difference of UBW-C compared with UBW-P and existing targeted backdoor watermarks lies in the generation of the modified subset $\mathcal{D}_m$. Specifically, in UBW-C, *we do not modify the labels of all poisoned samples, i.e., $\mathcal{D}_m = \{(\boldsymbol{x}', y) | \boldsymbol{x}' = G(\boldsymbol{x}; \boldsymbol{\theta}), (\boldsymbol{x}, y) \in \mathcal{D}_s\}$*. Before we reach the technical details of our UBW-C, we first present the necessary lemma and theorem.

**Lemma 1.** *The averaged class-wise dispersibility is always greater than or equal to the averaged sample-wise dispersibility i.e., $D_s \leq D_c$. Besides, the equality holds if and only if $f(\boldsymbol{x}_i) = f(\boldsymbol{x}_j), \forall i, j \in \{1, \cdots, N\}$.*

**Theorem 1.** *Let $f(\cdot; \boldsymbol{w})$ indicates the DNN with parameter $\boldsymbol{w}$, $G(\cdot; \boldsymbol{\theta})$ denotes the poisoned image generator with parameter $\boldsymbol{\theta}$, and $\mathcal{D} = \{(\boldsymbol{x}_i, y_i)\}_{i=1}^N$ is a given dataset with $K$ classes, we have*

$$\max_{\boldsymbol{\theta}} \sum_{i=1}^N H\left(f(G(\boldsymbol{x}_i; \boldsymbol{\theta}); \boldsymbol{w})\right) \leq \max_{\boldsymbol{\theta}} \sum_{j=1}^K \sum_{i=1}^N \mathbb{I}\{y_i = j\} \cdot H\left(\frac{\sum_{i=1}^N f(G(\boldsymbol{x}_i; \boldsymbol{\theta}); \boldsymbol{w}) \cdot \mathbb{I}\{y_i = j\}}{\sum_{i=1}^N \mathbb{I}\{y_i = j\}}\right).$$

Theorem 1 implies that *we can optimize the averaged sample-wise dispersibility $D_s$ and the class-wise dispersibility $D_c$ simultaneously by only maximizing $D_s$*. It motivates us to generate the modified subset $\mathcal{D}_m$ in our UBW-C (via optimizing generator $G$) as follows:

$$\max_{\boldsymbol{\theta}} \sum_{(\boldsymbol{x}, y) \in \mathcal{D}_s} \left[\mathcal{L}(f(G(\boldsymbol{x}; \boldsymbol{\theta}); \boldsymbol{w}^*), y) + \lambda \cdot H\left(f(G(\boldsymbol{x}; \boldsymbol{\theta}); \boldsymbol{w}^*)\right)\right], \tag{6}$$

$$s.t. \ \boldsymbol{w}^* = \arg\min_{\boldsymbol{w}} \sum_{(\boldsymbol{x}, y) \in \mathcal{D}_p} \mathcal{L}(f(\boldsymbol{x}; \boldsymbol{w}), y), \tag{7}$$

where $\lambda$ is a non-negative trade-off hyper-parameter.

In general, the aforementioned process is a standard bi-level optimization, which can be effectively and efficiently solved by alternatively optimizing the lower-level and upper-level sub-problems [44]. In particular, the optimization is conducted via stochastic gradient descent (SGD) with mini-batches [45], where estimating the class-wise dispersibility is difficult (especially when there are many classes). In contrast, *the estimation of sample-wise dispersibility $D_s$ is still simple and accurate even within a mini-batch*. It is another benefit of only using the averaged sample-wise dispersibility for optimization in our UBW-C. Please refer to the appendix for more our optimization details.

# 4 Towards Harmless Dataset Ownership Verification via UBW

## 4.1 Problem Formulation

Given a suspicious model, the defenders intend to verify whether it is trained on the (protected) dataset. Same as the previous work [3, 4], we assume that the dataset defenders can only query the suspicious model to obtain predicted probability vectors of input samples, whereas having no information about the training process and model parameters.

## 4.2 The Proposed Method

Since defenders can only modify the released dataset and query the suspicious model, the only way to tackle the aforementioned problem is to watermark the (unprotected) benign dataset so that models trained on it will have specific distinctive prediction behaviors. The dataset owners can release the watermarked dataset instead of the original one for copyright protection.

As described in Section 3, the DNNs watermarked by our UBW behave normally on benign samples while having dispersible predictions on poisoned samples. As such, it can be used to design harmless and stealthy dataset ownership verification. In general, given a suspicious model, the defenders can verify whether it was trained on the protected dataset by examining whether the model contains specific untargeted backdoor. *The model is regarded as trained on the protected dataset if it contains that backdoor.* To verify it, we design a hypothesis-test-based method, as follows:

**Proposition 1.** *Suppose $f(x)$ is the posterior probability of $x$ predicted by the suspicious model. Let variable $X$ denotes the benign sample and variable $X'$ is its poisoned version (i.e., $X' = G(X)$), while variable $P_b = f(X)_Y$ and $P_p = f(X')_Y$ indicate the predicted probability on the ground-truth label $Y$ of $X$ and $X'$, respectively. Given the null hypothesis $H_0 : P_b = P_p + \tau$ ($H_1 : P_b > P_p + \tau$) where the hyper-parameter $\tau \in [0, 1]$, we claim that the suspicious model is trained on the protected dataset (with $\tau$-certainty) if and only if $H_0$ is rejected.*

In practice, we randomly sample $m$ different benign samples to conduct the pairwise T-test [6] and calculate its p-value. The null hypothesis $H_0$ is rejected if the p-value is smaller than the significance level $\alpha$. In particular, *we only select samples that can be correctly classified by the suspicious model* to reduce the side-effects of model accuracy. Otherwise, due to the untargeted nature of our UBW, our verification may misjudge when there is dataset stealing, if the benign accuracy of the suspicious model is relatively low. Besides, we also calculate the *confidence score* $\Delta P = P_b - P_p$ to represent the verification confidence. *The larger the $\Delta P$, the more confident the verification.*

# 5 Experiments

## 5.1 Experimental Settings

**Datasets and Models.** In this paper, we conduct experiments on two classical benchmark datasets, including CIFAR-10 [1] and (a subset of) ImageNet [2], with ResNet-18 [46]. Specifically, we randomly select a subset containing 50 classes with $25,000$ images from the original ImageNet for training (500 images per class) and $2,500$ images for testing (50 images per class). For simplicity, all images are resized to $3 \times 64 \times 64$, following the settings used in Tiny-ImageNet [47].

**Baseline Selection.** We compare our UBW with representative existing poison-only backdoor attacks. Specifically, for attacks with poisoned labels, we adopt BadNets [30], blended attack (dubbed as 'Blended') [39], and WaNet [32] as the baseline methods. They are the representative of visible attacks, patch-based invisible attacks, and non-patch-based invisible attacks, respectively. We use the label-consistent attack (dubbed as 'Label-Consistent') [31] and Sleeper Agent [40] as the representative of attacks with clean labels. Besides, we also include the models trained on the benign dataset (dubbed as 'No Attack') as another baseline for reference.

## 5.2 The Performance of Dataset Watermarking

**Settings.** We set the poisoning rate $\gamma = 0.1$ for all watermarks on both datasets. In particular, since the label-consistent attack can only modify samples from the target class, its poisoning rate is set

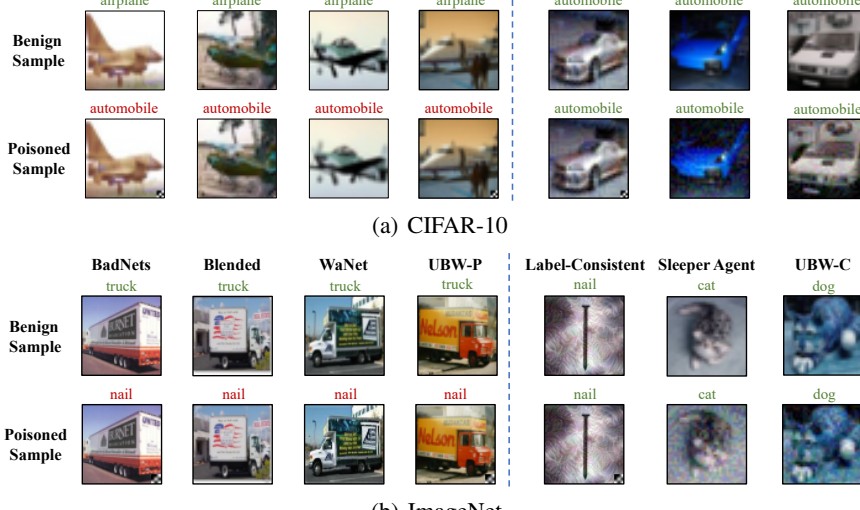

(a) CIFAR-10

(b) ImageNet

Figure 3: The example of samples involved in different backdoor watermarks. In the BadNets, blended attack, WaNet, and UBW-P, the labels of poisoned samples are inconsistent with their ground-truth ones. In the label-consistent attack, Sleeper Agent, and UBW-C, the labels of poisoned samples are the same as their ground-truth ones. In particular, the label-consistent attack can only poison samples in the target class, while other methods can modify all samples.

Table 1: The watermark performance on the CIFAR-10 dataset.

| Label Type↓ | Target Type↓ | Method↓, Metric→ | BA (%) | ASR-A (%) | ASR-C (%) | $D_p$ |
|---|---|---|---|---|---|---|
| N/A | | No Attack | 92.53 | N/A | N/A | N/A |
| Poisoned-Label | Targeted | BadNets | 91.52 | 100 | 100 | 0.0000 |
| | | Blended | 91.61 | 100 | 100 | 0.0000 |
| | | WaNet | 90.48 | 95.50 | 95.33 | 0.1979 |
| | Untargeted | UBW-P (Ours) | 90.59 | 92.30 | 92.51 | 2.2548 |
| Clean-Label | Targeted | Label-Consistent | 82.94 | 96.00 | 95.80 | 0.9280 |
| | | Sleeper Agent | 86.06 | 70.60 | 54.46 | 1.0082 |
| | Untargeted | UBW-C (Ours) | 86.99 | 89.80 | 87.56 | 1.2641 |

to its maximum (*i.e.*, 0.02) on the ImageNet dataset. The target label $y_t$ is set to 1 for all targeted watermarks. Besides, following the classical settings in existing papers, we adopt a white-black square as the trigger pattern for BadNets, blended attack, label-consistent attack, and UBW-P on both datasets. The trigger patterns adopted for Sleeper Agent and UBW-C are sample-specific. We set $\lambda = 2$ for UBW-C on both datasets. The example of poisoned samples generated by different methods is shown in Figure 3. More detailed settings are described in the appendix.

**Evaluation Metrics.** We use the benign accuracy (BA), the attack success rate (ASR), and the averaged prediction dispersibility ($D_p$) to evaluate the watermark performance. In particular, we introduce two types of ASR, including the attack success rate on all testing samples (ASR-A) and the attack success rate on correctly classified testing samples (ASR-C). In general, *the larger the BA, ASR, and $D_p$, the better the watermark*. Please refer to the appendix for more details.

**Results.** As shown in Table 1-2, *the performance of our UBW is on par with that of baseline targeted backdoor watermarks* under both poisoned-label and clean-label settings. Especially under the clean-label setting, our *UBW-C is significantly better than other watermarks with clean labels*. For example, the ASR-C increases of our method compared with label-consistent attack and Sleeper Agent are both over 55% on ImageNet. These results verify that our *UBW can implant distinctive behaviors in attacked DNNs*. In particular, our UBW has significantly higher averaged prediction dispersibility $D_p$, especially under the poisoned-label setting. For example, the $D_p$ of UBW-P is more than 10 times larger than that of all baseline attacks with poisoned labels on the CIFAR-10 dataset. These results verify that the *UBW can not manipulate malicious predictions deterministically and therefore is harmless*. Moreover, we notice that the $D_p$ of label-consistent attack and Sleeper Agent is similar to that of our UBW-C to some extent. It is mostly because targeted attacks with clean labels are significantly more difficult in making all poisoned samples to the same (target) class.

Table 2: The watermark performance on the ImageNet dataset.

| Label Type↓ | Target Type↓ | Method↓, Metric→ | BA (%) | ASR-A (%) | ASR-C (%) | $D_p$ |
|---|---|---|---|---|---|---|
| N/A | | No Attack | 67.30 | N/A | N/A | N/A |
| Poisoned-Label | Targeted | BadNets | 65.64 | 100 | 100 | 0.0000 |
| | | Blended | 65.28 | 88.00 | 85.37 | 0.3669 |
| | | WaNet | 62.56 | 78.00 | 73.17 | 0.7124 |
| | Untargeted | UBW-P (Ours) | 62.60 | 82.00 | 82.61 | 2.7156 |
| Clean-Label | Targeted | Label-Consistent | 62.36 | 30.00 | 2.78 | 1.2187 |
| | | Sleeper Agent | 56.92 | 6.00 | 2.31 | 1.0943 |
| | Untargeted | UBW-C (Ours) | 59.64 | 74.00 | 60.00 | 2.4010 |

Table 3: The effectiveness of dataset ownership verification via UBW-P.

| | CIFAR-10 | | | ImageNet | | |
|---|---|---|---|---|---|---|
| | Independent-T | Independent-M | Malicious | Independent-T | Independent-M | Malicious |
| $\Delta P$ | -0.0269 | 0.0024 | 0.7568 | 0.1281 | 0.0241 | 0.8000 |
| p-value | 1.0000 | 1.0000 | $10^{-36}$ | 0.9666 | 1.0000 | $10^{-10}$ |

Table 4: The effectiveness of dataset ownership verification via UBW-C.

| | CIFAR-10 | | | ImageNet | | |
|---|---|---|---|---|---|---|
| | Independent-T | Independent-M | Malicious | Independent-T | Independent-M | Malicious |
| $\Delta P$ | 0.1874 | 0.0171 | 0.6115 | 0.0588 | 0.1361 | 0.4836 |
| p-value | 0.9688 | 1.0000 | $10^{-14}$ | 0.9999 | 0.9556 | 0.0032 |

## 5.3 The Performance of UBW-based Dataset Ownership Verification

**Settings.** We evaluate our verification method in three representative scenarios, including **1)** independent trigger (dubbed as 'Independent-T'), **2)** independent model (dubbed as 'Independent-M'), and **3)** unauthorized dataset usage (dubbed as 'Malicious'). In the first scenario, we query the attacked suspicious model using the trigger that is different from the one used for model training; In the second scenario, we examine the benign suspicious model using the trigger pattern; We adopt the trigger used in the training process of the watermarked suspicious model in the last scenario. We set $\tau = 0.25$ for the hypothesis-test in all cases. More detailed settings are in the appendix.

**Evaluation Metrics.** We adopt the $\Delta P \in [-1, 1]$ and the p-value $\in [0, 1]$ for the evaluation. For the two independent scenarios, the smaller the $\Delta P$ and the larger the p-value, the better the verification; For the malicious one, the larger the $\Delta P$ and the smaller the p-value, the better the verification.

**Results.** As shown in Table 3-4, our dataset ownership verification is effective in all cases, no matter under UBW-P or UBW-C. Specifically, our method can accurately identify unauthorized dataset usage (*i.e.*, 'Malicious') with high confidence (*i.e.*, $\Delta P \gg 0$ and p-value $\ll 0.01$) while does not misjudge (*i.e.*, $\Delta P$ is nearly 0 and p-value $\gg 0.05$) when there is no stealing (*i.e.*, 'Independent-T' and 'Independent-M'). For example, the p-values of verifying independent cases are all nearly 1 on both datasets. We notice that the verification performance under UBW-C is relatively poorer than that under UBW-P, although its performance is already capable enough for verification. However, the UBW-C is more stealthy, since the labels of poisoned samples are consistent with their ground-truth label and the trigger patterns are invisible. Users can adopt different UBWs based on their needs.

## 5.4 Discussion

### 5.4.1 The Ablation Study

In this section, we explore the effects of key hyper-parameters involved in our UBW. The detailed settings and the effects of hyper-parameters involved in ownership verification are in the appendix.

**Effects of Poisoning Rate $\gamma$.** As shown in Figure 4, the attack success rate (ASR) increases with the increase of the poisoning rate $\gamma$. Both UBW-P and UBW-C reach promising ASR even when $\gamma$ is small (*e.g.*, 0.03). Besides, the benign accuracy decreases with the increase of $\gamma$. Users should assign $\gamma$ based on their specific requirements in practice.

**Effects of Trade-off Hyper-parameter $\lambda$.** As shown in Figure 5, the averaged prediction dispersibility $D_p$ increases with the increase of $\lambda$. This phenomenon indicates that the averaged sample-wise dispersibility $D_s$ used in our UBW-C is a good approximation of $D_p$. In contrast, increasing $\lambda$ has minor effects in ASR, which is probably because the untargeted attack scheme is more stable.

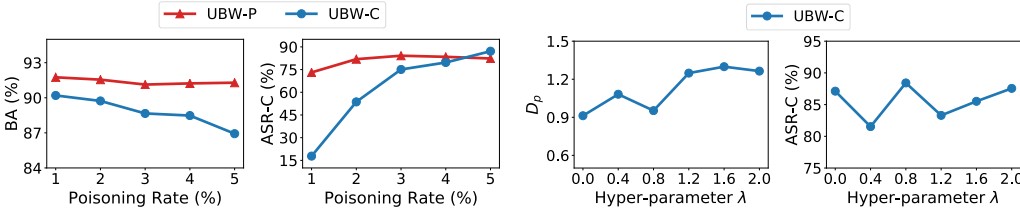

Figure 4: The effects of poisoning rate $\gamma$.     Figure 5: The effects of hyper-parameter $\lambda$.

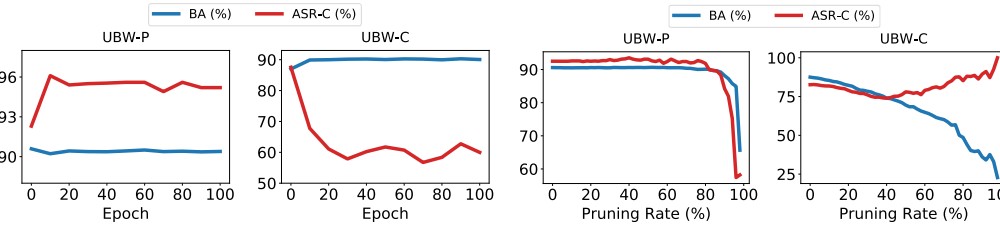

Figure 6: The resistance to fine-tuning.     Figure 7: The resistance to model pruning.

### 5.4.2 Resistance to Backdoor Defenses

In this section, we discuss whether our UBW is resistant to existing backdoor defenses so that it can still provide promising dataset protection even under adaptive opposite methods. In particular, the trigger patterns used by our UBW-C are *sample-specific*, where different poisoned images contain different triggers (as shown in Figure 3). Recently, ISSBA [42] revealed that most of the existing defenses (*e.g.*, Neural Cleanse [48], SentiNet [49], and STRIP [50]) have a latent assumption that the trigger patterns are *sample-agnostic*. Accordingly, our *UBW-C can naturally bypass them*, since it breaks their fundamental assumption. Here we explore the resistance of our UBW to fine-tuning [51, 52] and model pruning [52, 53], which are the representative defenses whose effects did not rely on this assumption. The detailed settings and resistance to other defenses are in the appendix.

As shown in Figure 6, our *UBW is resistant to fine-tuning*. Specifically, the attack success rates are still larger than 55% for both UBW-P and UBW-C after the fine-tuning process is finished. Besides, our *UBW is also resistant to model pruning* (as shown in Figure 7). The ASRs of both UBW-P and UBW-C are larger than 50% even under high pruning rates, where the benign accuracies are already low. An interesting phenomenon is that as the pruning rate increases, the ASR of UBW-C even increases for a period. We speculate that it is probably because our UBW-C is untargeted and sample-specific, and therefore it can reach better attack effects when the model's benign functions are significantly depressed. We will further discuss its mechanism in our future work.

## 6 Societal Impacts

This paper is the first attempt toward untargeted backdoor attacks and their positive applications. In general, our main focus is how to design and use untargeted backdoor attacks as harmless and stealthy watermarks for dataset protection, which has positive societal impacts. We notice that our untargeted backdoor watermark (UBW) is resistant to existing backdoor defenses and could be maliciously used by the backdoor adversaries. However, compared with existing targeted backdoor attacks, our UBW is untargeted and therefore has minor threats. Moreover, although an effective defense is yet to be developed, people can still mitigate or even avoid the threats by only using trusted training resources.

## 7 Conclusion

In this paper, we revisited how to protect the copyrights of (open-sourced) datasets. We revealed that existing dataset ownership verification could introduce new serious risks, due to the targeted nature of existing poison-only backdoor attacks used for dataset watermarking. Based on this understanding, we explored the untargeted backdoor watermark (UBW) paradigm under both poisoned-label and clean-label settings, whose abnormal model behaviors were not deterministic. We also studied how to exploit our UBW for harmless and stealthy dataset ownership verification. Experiments on benchmark datasets validated the effectiveness of our method and its resistance to backdoor defenses.

## Acknowledgments

This work is supported in part by the National Natural Science Foundation of China under Grant 62171248, the PCNL Key Project (PCL2021A07), the Tencent Rhino-Bird Research Program, and the C3 AI and Amazon research awards. We also sincerely thank Linghui Zhu from Tsinghua University for her assistance in the experiments of resistance to saliency-based backdoor defenses.

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
