# Appendix:
# Untargeted Backdoor Watermark: Towards Harmless and Stealthy Dataset Copyright Protection

## A  The Omitted Proofs

**Lemma 1.** *The averaged class-wise dispersibility is always greater than or equal to the averaged sample-wise dispersibility i.e., $D_s \leq D_c$. Besides, the equality holds if and only if $f(\boldsymbol{x}_i) = f(\boldsymbol{x}_j), \forall i, j \in \{1, \cdots, N\}$.*

*Proof.* Since entropy is a concave function [1], according to Jensen's inequality, we have:

$$H\left(\frac{\sum_{i=1}^{N} f(\boldsymbol{x}_i) \cdot \mathbb{I}\{y_i = j\}}{\sum_{i=1}^{N} \mathbb{I}\{y_i = j\}}\right) \geq \sum_{i=1}^{N} \frac{\mathbb{I}\{y_i = j\}}{\sum_{i=1}^{N} \mathbb{I}\{y_i = j\}} H\left(f(\boldsymbol{x}_i)\right) = H\left(f(\boldsymbol{x}_i)\right). \tag{1}$$

Since each sample $\boldsymbol{x}$ has and only has one label $y \in \{1, \cdots, K\}$, we have:

$$H\left(f(\boldsymbol{x}_i)\right) = \sum_{j=1}^{K} H\left(f(\boldsymbol{x}_i)\right) \cdot \mathbb{I}\{y_i = j\}, \forall i \in \{1, \cdots, N\}. \tag{2}$$

As such,

$$D_c \geq \frac{1}{N} \sum_{j=1}^{K} \sum_{i=1}^{N} \mathbb{I}\{y_i = j\} \cdot H\left(f(\boldsymbol{x}_i)\right) = \frac{1}{N} \sum_{i=1}^{N} H\left(f(\boldsymbol{x}_i)\right) \triangleq D_s. \tag{3}$$

Moreover, since entropy is strictly concave (*i.e.*, non-linear) [1], in equation (1)&(3), the equality holds if and only if $f(\boldsymbol{x}_i) = f(\boldsymbol{x}_j), \forall i, j \in \{1, \cdots, N\}$.

$\square$

**Theorem 1.** *Let $f(\cdot; \boldsymbol{w})$ indicates the DNN with parameter $\boldsymbol{w}$, $G(\cdot; \boldsymbol{\theta})$ denotes the poisoned image generator with parameter $\boldsymbol{\theta}$, and $\mathcal{D} = \{(\boldsymbol{x}_i, y_i)\}_{i=1}^{N}$ is a given dataset with $K$ classes, we have*

$$\max_{\boldsymbol{\theta}} \sum_{i=1}^{N} H\left(f(G(\boldsymbol{x}_i; \boldsymbol{\theta}); \boldsymbol{w})\right) \leq \max_{\boldsymbol{\theta}} \sum_{j=1}^{K} \sum_{i=1}^{N} \mathbb{I}\{y_i = j\} \cdot H\left(\frac{\sum_{i=1}^{N} f(G(\boldsymbol{x}_i; \boldsymbol{\theta}); \boldsymbol{w}) \cdot \mathbb{I}\{y_i = j\}}{\sum_{i=1}^{N} \mathbb{I}\{y_i = j\}}\right).$$

*Proof.* The proof is straightforward given Lemma 1, based on replacing $f(\boldsymbol{x}_i)$ with $f(G(\boldsymbol{x}_i; \boldsymbol{\theta}); \boldsymbol{w})$ and maximizing both sides simultaneously.

$\square$

## B The Optimization Process of our UBW-C

Recall that the optimization objective of our UBW-C is as follows:

$$\max_{\boldsymbol{\theta}} \sum_{(\boldsymbol{x},y)\in\mathcal{D}_s} \left[\mathcal{L}(f(G(\boldsymbol{x};\boldsymbol{\theta});\boldsymbol{w}^*),y) + \lambda \cdot H\left(f(G(\boldsymbol{x};\boldsymbol{\theta});\boldsymbol{w}^*)\right)\right], \tag{4}$$

$$s.t.\ \boldsymbol{w}^* = \arg\min_{\boldsymbol{w}} \sum_{(\boldsymbol{x},y)\in\mathcal{D}_p} \mathcal{L}(f(\boldsymbol{x};\boldsymbol{w}),y), \tag{5}$$

where $\lambda$ is a non-negative trade-off hyper-parameter.

In general, the aforementioned process is a standard bi-level optimization, which can be effectively and efficiently solved by alternatively optimizing the upper-level and lower-level sub-problems [2]. To solve the aforementioned problem, the form of $G$ is one of the key factors. Inspired by the hidden trigger backdoor attack [3] and the Sleeper Agent [4], we also adopt different generators during the training and inference process to enhance attack effectiveness and stealthiness, as follows:

Let $G_t$ and $G_i$ denote the generator used in the training and inference process, respectively. We intend to generate *sample-specific* small additive perturbations for selected training images based on $G_t$ so that their gradient ensemble has a similar direction to the gradient ensemble of poisoned 'testing' images generated by $G_i$. Specifically, we set $G_t(\boldsymbol{x}) = \boldsymbol{x} + \boldsymbol{\theta}(\boldsymbol{x})$, where $||\boldsymbol{\theta}(\boldsymbol{x})||_\infty \leq \epsilon$ and $\epsilon$ is the perturbation budget; We set $G_i(\boldsymbol{x}) = (\boldsymbol{1} - \boldsymbol{\alpha}) \otimes \boldsymbol{x} + \boldsymbol{\alpha} \otimes \boldsymbol{t}$, where $\boldsymbol{\alpha} \in \{0,1\}^{C \times W \times H}$ denotes the given mask and $\boldsymbol{t} \in \mathcal{X}$ is the given trigger pattern. In general, the trigger patterns used for training is invisible for stealthiness while those used for inference is visible for effectiveness. The detailed lower-level and upper-level sub-problems are as follows:

**Upper-level Sub-problem.** Given the current model parameters $\boldsymbol{w}$, we optimize the trigger patterns $\{\boldsymbol{\theta}(\boldsymbol{x})|\boldsymbol{x} \in \mathcal{D}_s\}$ of selected training samples (for poisoning) based on the gradient matching:

$$\max_{\{\boldsymbol{\theta}(\boldsymbol{x})|\boldsymbol{x}\in\mathcal{D}_s,\ ||\boldsymbol{\theta}(\boldsymbol{x})||_\infty \leq \epsilon\}} \frac{\nabla_{\boldsymbol{w}}\mathcal{L}_t \cdot \nabla_{\boldsymbol{w}}\mathcal{L}_i}{||\mathcal{L}_t|| \cdot ||\mathcal{L}_i||}, \tag{6}$$

where

$$\mathcal{L}_i = \frac{1}{N} \cdot \sum_{(\boldsymbol{x},y)\in\mathcal{D}} \left[\mathcal{L}(f(G_i(\boldsymbol{x});\boldsymbol{w}),y) + \lambda \cdot H\left(f(G_i(\boldsymbol{x});\boldsymbol{w})\right)\right], \tag{7}$$

$$\mathcal{L}_t = \frac{1}{M} \cdot \sum_{(\boldsymbol{x},y)\in\mathcal{D}_s} \mathcal{L}(f(\boldsymbol{x} + \boldsymbol{\theta}(\boldsymbol{x});\boldsymbol{w}),y), \tag{8}$$

$N$ and $M$ denote the number of training samples and the number of selected samples, respectively. The upper-level sub-problem is solved by projected gradient ascend (PGA) [5].

**Lower-level Sub-problem.** Given the current trigger patterns $\{\boldsymbol{\theta}(\boldsymbol{x})|\boldsymbol{x} \in \mathcal{D}_s\}$, we can obtain the poisoned training dataset $\mathcal{D}_p$ and then optimize the model parameters $\boldsymbol{w}$ via

$$\min_{\boldsymbol{w}} \sum_{(\boldsymbol{x},y)\in\mathcal{D}_p} \mathcal{L}(f(\boldsymbol{x};\boldsymbol{w}),y). \tag{9}$$

The lower-level sub-problem is solved by stochastic gradient descent (SGD) [5].

Besides, there are three additional optimization details that we need to mention, as follows:

**1) How to Select Training Samples for Poisoning.** We select training samples with the largest gradient norms instead of random selection for poisoning since they have more influence. It is allowed in our UBW since the dataset owner can determine which samples should be modified.

**2) How to Select 'Test' Samples for Poisoning.** Instead of using all training samples to calculate Eq. (7), we only use those from a specific source class. This approach is used to further enhance UBW effectiveness, since the gradient ensemble of samples from all classes may be too 'noisy' to learn for $G_t$. Its benefits are verified in the following Section F.

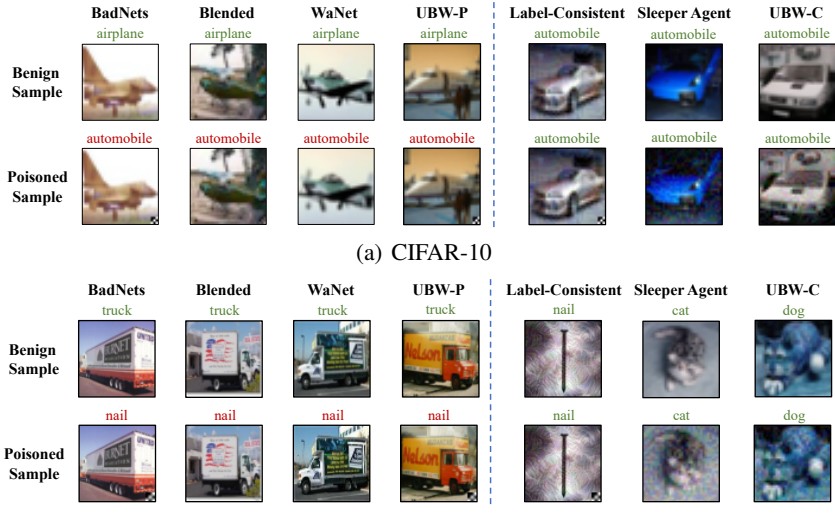

(a) CIFAR-10

(b) ImageNet

Figure 1: The example of samples involved in different backdoor watermarks. In the BadNets, blended attack, WaNet, and UBW-P, the labels of poisoned samples are inconsistent with their ground-truth ones. In the label-consistent attack, Sleeper Agent, and UBW-C, the labels of poisoned samples are the same as their ground-truth ones. In particular, the label-consistent attack can only poison samples in the target class, while other methods can modify all samples.

**3) The Relation between Dispersibility and Attack Success Rate.** In general, the optimization of dispersibility contradicts to that of the attack success rate to some extent. Specifically, let us consider a classification problem with $K$ different classes. When the averaged sample-wise dispersibility used in optimizing UBW-C reaches its maximum value, the attack success rate is only $\frac{K-1}{K}$, since the predicted probability vectors are all uniform; When the attack success rate reaches 100%, both averaged prediction dispersibility and sample-wise dispersibility cannot reach their maximum.

In particular, similar to other backdoor attacks based on bi-level optimization (*e.g.*, LIRA [6] and Sleeper Agent [4]), we notice that the watermark performance of our UBW-C is not very stable across different random seeds (*i.e.*, has relatively large standard deviation). We will explore how to stabilize and improve the performance of UBW-C in our future work.

# C    Detailed Experimental Settings

## C.1    Detailed Settings for Dataset Watermarking

**Datasets and Models.** In this paper, we conduct experiments on two classical benchmark datasets, including CIFAR-10 [7] and (a subset of) ImageNet [8], with ResNet-18 [9]. Specifically, we randomly select a subset containing 50 classes with $25,000$ images from the original ImageNet for training (500 images per class) and $2,500$ images for testing (50 images per class). For simplicity, all images are resized to $3 \times 64 \times 64$, following the settings used in Tiny-ImageNet [10].

**Baseline Selection.** We compare our UBW with representative existing poison-only backdoor attacks. Specifically, for attacks with poisoned labels, we adopt BadNets [11], blended attack (dubbed as 'Blended') [12], and WaNet [13] as the baseline methods. They are the representative of visible attacks, patch-based invisible attacks, and non-patch-based invisible attacks, respectively. We use the label-consistent attack (dubbed as 'Label-Consistent') [14] and Sleeper Agent [4] as the representative of attacks with clean labels. Besides, we also include the models trained on the benign dataset (dubbed as 'No Attack') as another baseline for reference.

**Attack Setup.** We implement BadNets, blended attack, and label-consistent attack based on the open-sourced Python toolbox—BackdoorBox [15]. The experiments of Sleeper Agent are conducted based on its official open-sourced codes[1]. We set the poisoning rate $\gamma = 0.1$ for all attacks on both datasets. In particular, since the label-consistent attack can only modify samples from the target class, its poisoning rate is set to its maximum (*i.e.*, 0.02) on the ImageNet dataset. The target label

---

[1]`https://github.com/hsouri/Sleeper-Agent`

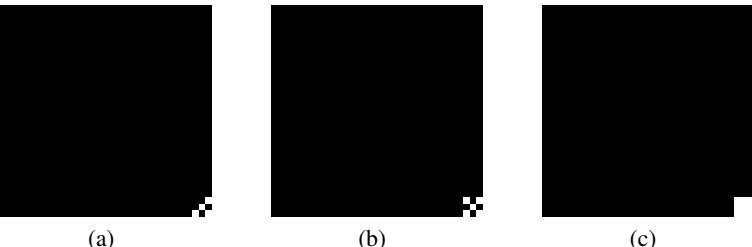

(a)          (b)          (c)

Figure 2: The trigger patterns used for evaluation.

$y_t$ is set to 1 for all targeted attacks. Besides, following the classical settings in existing papers, we adopt a white-black square as the trigger pattern for BadNets, blended attack, label-consistent attack, and UBW-P on both datasets. The trigger patterns adopted for training Sleeper Agent and UBW-C are sample-specific, while those used in the inference process are the same as those used by BadNets, blended attack, label-consistent attack, and UBW-P. Specifically, for the blended attack, the blended ratio $\alpha$ is set to 0.1; For the label-consistent attack, we used the projected gradient descent (PGD) [16] to generate adversarial perturbations within the $\ell^\infty$-ball for pre-processing selected images before the poisoning, where the maximum perturbation size $\epsilon = 16$, step size 1.5, and 30 steps. For the WaNet, we adopted its default settings provided by `BackdoorBox` with noise mode. For both Sleeper Agent and our UBW-C, we alternatively optimize the upper-level and lower-level sub-problems 5 times, where we train the model 50 epochs and generate the trigger patterns with PGA-40 on the CIFAR-10 dataset. On the ImageNet dataset, we alternatively optimize the upper-level and lower-level sub-problems 3 times, where we train the model 40 epochs and generate the trigger patterns via PGA-30. The initial model parameters are obtained by training on the benign dataset. We set $\lambda = 2$ and the source class is set as 0 on both datasets. The example of poisoned training samples generated by different attacks is shown in Figure 1.

**Training Setup.** On both CIFAR-10 and ImageNet datasets, we train the model 200 epochs with batch size 128. Specifically, we use the SGD optimizer with a momentum of 0.9, weight decay of $5 \times 10^{-4}$, and an initial learning rate of 0.1. The learning rate is decreased by a factor of 10 at the epoch of 150 and 180, respectively. In particular, we add trigger patterns before performing the data augmentation with horizontal flipping.

## C.2 Detailed Settings for Dataset Ownership Verification

We evaluate our verification method in three representative scenarios, including **1)** independent trigger (dubbed as 'Independent-T'), **2)** independent model (dubbed as 'Independent-M'), and **3)** unauthorized dataset usage (dubbed as 'Malicious'). In the first scenario, we query the attacked suspicious model using the trigger that is different from the one used for model training; In the second scenario, we examine the benign suspicious model using the trigger pattern; We adopt the trigger used in the training process of the watermarked suspicious model in the last scenario. Moreover, we sample $m = 100$ samples on CIFAR10 and $m = 30$ samples on ImageNet and set $\tau = 0.25$ for the hypothesis-test in each case for both UBW-P and UBW-C. We use $m = 30$ on ImageNet since there is only 50 testing images from the source class and we only select samples that can be correctly classified by the suspicious model to reduce the side-effects of model accuracy.

## C.3 Detailed Settings for Resistance to Backdoor Defenses

**Settings for Fine-tuning.** We conduct the experiments on the CIFAR-10 dataset as an example for discussion. Following its default settings, we freeze the convolutional layers and tune the remaining fully-connected layers of the watermarked DNNs. Specifically, we adopt 10% benign training samples for fine-tuning and set the learning rate as 0.1. We fine-tune the model 100 epochs in total.

**Settings for Model Pruning.** We conduct the experiments on the CIFAR-10 dataset as an example for discussion. Following its default settings, we conduct channel pruning [17] on the output of the last convolutional layer with 10% benign training samples. The pruning rate $\beta \in \{0\%, 2\%, \cdots, 98\%\}$.

Table 1: The effectiveness of our UBW with different trigger patterns on the CIFAR-10 dataset.

| Method↓ | Pattern↓, Metric→ | BA (%) | ASR-A (%) | ASR-C (%) | $D_p$ |
|---------|-------------------|--------|-----------|-----------|-------|
| UBW-P | Pattern (a) | 90.59 | 92.30 | 92.51 | 2.2548 |
| | Pattern (b) | 90.31 | 84.53 | 82.39 | 2.2331 |
| | Pattern (c) | 90.21 | 87.78 | 86.94 | 2.2611 |
| UBW-C | Pattern (a) | 86.99 | 89.80 | 87.56 | 1.2641 |
| | Pattern (b) | 86.25 | 90.90 | 88.91 | 1.1131 |
| | Pattern (c) | 87.78 | 81.23 | 78.55 | 1.0089 |

Table 2: The effectiveness of our UBW with different trigger sizes on the CIFAR-10 dataset.

| Method↓ | Trigger Size↓, Metric→ | BA (%) | ASR-A (%) | ASR-C (%) | $D_p$ |
|---------|------------------------|--------|-----------|-----------|-------|
| UBW-P | 2 | 90.55 | 82.60 | 82.21 | 2.2370 |
| | 4 | 90.37 | 83.50 | 83.30 | 2.2321 |
| | 6 | 90.43 | 86.30 | 86.70 | 2.2546 |
| | 8 | 90.46 | 86.40 | 86.26 | 2.2688 |
| | 10 | 90.72 | 86.10 | 85.82 | 2.2761 |
| | 12 | 90.22 | 88.30 | 87.94 | 2.2545 |
| UBW-C | 2 | 87.34 | 4.38 | 15.00 | 0.7065 |
| | 4 | 87.71 | 70.80 | 64.86 | 1.2924 |
| | 6 | 87.69 | 75.60 | 70.85 | 1.7892 |
| | 8 | 88.89 | 75.40 | 69.86 | 1.2904 |
| | 10 | 88.30 | 77.60 | 73.92 | 1.7534 |
| | 12 | 89.29 | 98.00 | 97.72 | 1.1049 |

## D   The Effects of Trigger Patterns and Sizes

### D.1   The Effects of Trigger Patterns

**Settings.** In this section, we conduct experiments on the CIFAR-10 dataset to discuss the effects of trigger patterns. Except for the trigger pattern, all other settings are the same as those used in Section C.1. The adopted trigger patterns are shown in Figure 2.

**Results.** As shown in Table 1, both UBW-P and UBW-C are effective with each trigger pattern, although the performance may have some fluctuations. Specifically, the ASR-As are larger than 80% in all cases. These results verify that both UBW-P and UBW-C can reach promising performance with arbitrary user-specified trigger patterns used in the inference process.

### D.2   The Effects of Trigger Sizes

**Settings.** In this section, we conduct experiments on the CIFAR-10 dataset to discuss the effects of trigger sizes. Except for the trigger size, all other settings are the same as those used in Section C.1. The specific trigger patterns are generated based on resizing the one used in our main experiments.

**Results.** As shown in Table 2, the attack success rate increases with the increase of trigger size. In particular, different from existing (targeted) patch-based backdoor attacks (*e.g.*, BadNets and blended attack), increasing the trigger size has minor adverse effects in reducing the benign accuracy, which is most probably due to our untargeted attack paradigm. The benign accuracy even slightly increases with the increase of trigger sizes on UBW-C, which is mostly because the trigger pattern is not directly added to the poisoned samples during the training process (as described in Section B).

## E   The Effects of Verification Certainty and Number of Sampled Images

### E.1   The Effects of Verification Certainty

**Settings.** In this section, we conduct experiments on the CIFAR-10 dataset to discuss the effects of verification certainty $\tau$ in UBW-based dataset ownership verification. Except for the $\tau$, all other settings are the same as those used in Section C.2.

Table 3: The p-value of UBW-based dataset ownership verification $w.r.t.$ the verification certainty $\tau$ on the CIFAR-10 dataset.

| Method↓ | Scenario↓, $\tau\rightarrow$ | 0 | 0.05 | 0.1 | 0.15 | 0.2 | 0.25 |
|---|---|---|---|---|---|---|---|
| UBW-P | Independent-T | 0.1705 | 1.0 | 1.0 | 1.0 | 1.0 | 1.0 |
| | Independent-M | 0.2178 | 1.0 | 1.0 | 1.0 | 1.0 | 1.0 |
| | Malicious | $10^{-51}$ | $10^{-48}$ | $10^{-45}$ | $10^{-42}$ | $10^{-39}$ | $10^{-36}$ |
| UBW-C | Independent-T | $10^{-8}$ | $10^{-5}$ | 0.0049 | 0.1313 | 0.6473 | 0.9688 |
| | Independent-M | 0.1821 | 0.9835 | 1.0 | 1.0 | 1.0 | 1.0 |
| | Malicious | $10^{-27}$ | $10^{-24}$ | $10^{-22}$ | $10^{-19}$ | $10^{-16}$ | $10^{-14}$ |

Table 4: The p-value of UBW-based dataset ownership verification $w.r.t.$ the number of sampled images $m$ on the CIFAR-10 dataset.

| Method↓ | Scenario↓, $m\rightarrow$ | 20 | 40 | 60 | 80 | 100 | 120 |
|---|---|---|---|---|---|---|---|
| UBW-P | Independent-T | 1.0 | 1.0 | 1.0 | 1.0 | 1.0 | 1.0 |
| | Independent-M | 1.0 | 1.0 | 1.0 | 1.0 | 1.0 | 1.0 |
| | Malicious | $10^{-7}$ | $10^{-14}$ | $10^{-23}$ | $10^{-32}$ | $10^{-36}$ | $10^{-42}$ |
| UBW-C | Independent-T | 0.9348 | 0.9219 | 0.9075 | 0.9093 | 0.9688 | 0.9770 |
| | Independent-M | 1.0 | 1.0 | 1.0 | 1.0 | 1.0 | 1.0 |
| | Malicious | $10^{-3}$ | $10^{-6}$ | $10^{-7}$ | $10^{-10}$ | $10^{-14}$ | $10^{-16}$ |

Table 5: The effectiveness of UBW-C when attacking all samples or samples from the source class.

| Dataset↓ | Scenario↓, Metric→ | BA (%) | ASR-A (%) | ASR-C (%) | $D_p$ |
|---|---|---|---|---|---|
| CIFAR-10 | All | **87.42** | 58.83 | 50.31 | 0.9843 |
| | Source (Ours) | 86.99 | **89.80** | **87.56** | **1.2641** |
| ImageNet | All | 58.64 | 42.03 | 21.27 | 2.1407 |
| | Source (Ours) | **59.64** | **74.00** | **60.00** | **2.4010** |

**Results.** As shown in Table 3, the p-value increases with the increase of verification certainty $\tau$ in all scenarios. In particular, when $\tau$ is smaller than 0.15, UBW-C will misjudge the cases of Independent-T. This failure is due to the untargeted nature of our UBW and why we introduced $\tau$ in our verification process. Besides, the larger the $\tau$, the unlikely the misjudgments happen and the more likely that the dataset stealing is ignored. People should assign $\tau$ based on their specific needs.

**E.2   The Effects of the Number of Sampled Images**

**Settings.** In this section, we conduct experiments on the CIFAR-10 dataset to study the number of sampled images $m$ in UBW-based dataset ownership verification. Except for the $m$, all other settings are the same as those used in Section C.2.

**Results.** As shown in Table 4, the p-value decreases with the increase of $m$ in the malicious scenario while it decreases with the increase of $m$ in the independent scenarios. In other words, the probability that our UBW-based dataset ownership verification makes correct judgments increases with the increase of $m$. This benefit is mostly because increasing $m$ will reduce the adverse effects of the randomness involved in the sample selection.

**F   The Effectiveness of UBW-C When Attacking All Classes**

As described in Section B, our UBW-C randomly selects samples from a random source class instead of all classes for gradient matching. This special design is to reduce the optimization difficulty, since the gradient ensemble of samples from different classes may be too 'noisy' to learn for the poisoned training image generator $G_t$. In this section, we verify its effectiveness by comparing our UBW-C with its variant, which uses all samples for gradient matching.

As shown in Table 5, only using source class samples is significantly better than using all samples during the optimization of UBW-C. Specifically, the ASR-A increases of UBW-C compared with its variant are larger than 30% on both CIFAR-10 and ImageNet. Besides, we notice that the averaged

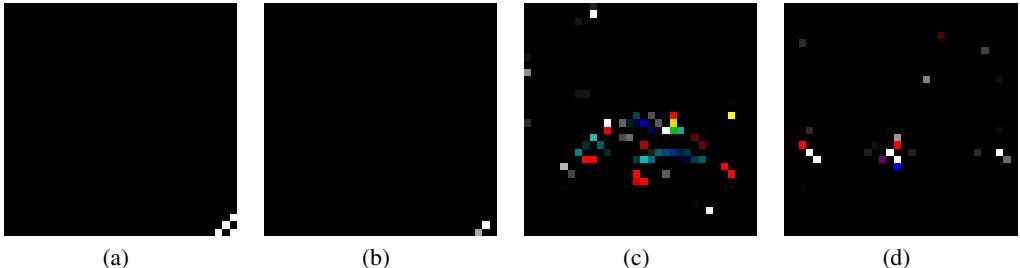

|     (a)     |     (b)     |     (c)     |     (d)     |

Figure 3: The ground-truth trigger pattern and those synthesized by neural cleanse. **(a)**: ground-truth trigger pattern; **(b)**: synthesized trigger pattern of BadNets; **(c)**: synthesized trigger pattern of UBW-P; **(d)**: synthesized trigger pattern of UBW-C. The synthesized pattern of BadNets is similar to the ground-truth one whereas those of our UBW-P and UBW-C are meaningless.

prediction dispersibility $D_p$ of the UBW-C variant is similar to that of our UBW-C to some extent. It is mostly because our UBW-C is untargeted and the variant has relatively low benign accuracy.

# G    Resistance to Other Backdoor Defenses

In this section, we discuss the resistance of our UBW-P and UBW-C to more potential backdoor defenses. We conduct experiments on the CIFAR-10 dataset as an example for the discussion.

## G.1    Resistance to Trigger Synthesis based Defenses

Currently, there are many trigger synthesis based backdoor defenses [18, 19, 20], which synthesized the trigger pattern for backdoor unlearning or detection. Specifically, they first generate the potential trigger pattern for each class and then filter the final synthetic one based on anomaly detection. In this section, we verify that our UBW can bypass these defenses for it breaks their latent assumption that the backdoor attacks are targeted.

**Settings.** Since neural cleanse [18] is the first and the most representative trigger synthesis based defense, we adopt it as an example to synthesize the trigger pattern of DNNs watermarked by BadNets and our UBW-P and UBW-C. We implement it based on its open-sourced codes[2] and default settings.

**Results.** As shown in Figure 3, the synthesized pattern of BadNets is similar to the ground-truth trigger pattern. However, those of our UBW-P and UBW-C are significantly different from the ground-truth one. These results show that our UBW is resistant to trigger synthesis based defenses.

## G.2    Resistance to Saliency-based Defenses

Since the attack effectiveness is mostly caused by the trigger pattern, there were also some backdoor defenses [21, 22, 23] based on detecting trigger areas with saliency maps. Specifically, these methods first generated the saliency map of each sample and then obtained trigger regions based on the intersection of all generated saliency maps. Since our UBW is untargeted, the relation between the trigger pattern and the predicted label is less significant compared with existing targeted backdoor attacks. As such, it can bypass those saliency-based defenses, which is verified in this section.

**Settings.** We generate the saliency maps of models watermarked by BadNets and our UBW-P and UBW-C, based on the Grad-CAM [24] with its default settings. We randomly select samples from the source class to generate their poisoned version for the discussion.

**Results.** As shown in Figure 4, the Grad-CAM mainly focuses on the trigger areas of poisoned images in BadNets. In contrast, it mainly focuses on other regions (*e.g.*, object outline) of poisoned images in our UBW-C. We notice that the Grad-CAM also focuses on the trigger areas in our UBW-P in a few cases. It is most probably because the trigger pattern used in the inference process is the

---

[2]https://github.com/bolunwang/backdoor

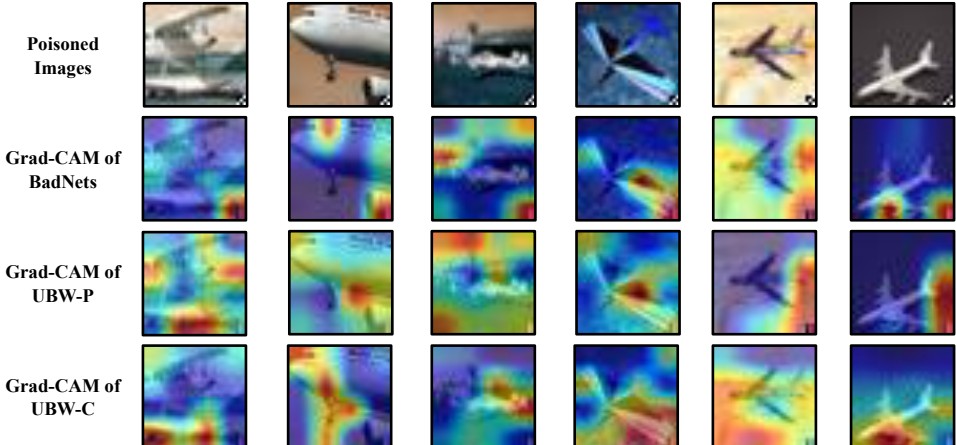

Figure 4: The poisoned images and their saliency maps based on Grad-CAM with DNNs watermarked by different methods. The Grad-CAM mainly focuses on the trigger areas of poisoned images in BadNets, while it mainly focuses on other regions (*e.g.*, object outline) in our UBW.

Table 6: The averaged entropy generated by STRIP of models watermarked by different methods. The larger the entropy, the harder for STRIP to detect the watermark.

| Metric↓, Method→ | BadNets | UBW-P | UBW-C |
|---|---|---|---|
| Averaged Entropy | 0.0093 | 1.5417 | 1.2018 |

same as the one used for training in our UBW-P while we use invisible additive noises in our training process of UBW-C. These results validate that our UBW is resistant to saliency-based defenses.

### G.3 Resistance to STRIP

Recently, Gao *et al*. [25] proposed STRIP to filter poisoned samples based on the prediction variation of samples generated by imposing various image patterns on the suspicious image. The variation is measured by the entropy of the average prediction of those samples. Specifically, the STRIP assumed that the trigger pattern is sample-agnostic and the attack is targeted. Accordingly, the more likely the suspicious image contains trigger pattern, the smaller the entropy since those modified images will still be predicted as the target label so that the average prediction is still nearly an one-hot vector.

**Settings.** We randomly select 100 testing images from the source class to generate their poisoned version, based on BadNets and our UBW-P and UBW-C. We calculate the entropy of each poisoned image based on the open-sourced codes[3] and default settings of STRIP. We then calculate the averaged entropy among all poisoned samples for each watermarking method as their indicator. The larger the entropy, the harder for STRIP to detect the watermark.

**Results.** As shown in Table 6, the averaged entropies of both UBW-P and UBW-C are significantly higher than that of BadNets. Specifically, the entropies of both UBW-P and UBW-C are more than 100 times larger than that of BadNets. It is mostly due to the untargeted nature of our UBW whose predictions are dispersed. These results verify that our UBW is resistant to STRIP.

### G.4 Resistance to Dataset-level Backdoor Defenses

In this section, we discuss whether our methods are resistant to dataset-level backdoor defenses.

**Settings.** In this part, we adopt the spectral signatures [26] and the activation clustering [27] as representative dataset-level backdoor defenses for our discussion. Both spectral signatures and activation clustering tend to filter poisoned samples from the training dataset, based on sample behaviors in hidden feature space. We implement these methods based on their official open-sourced

---

[3]`https://github.com/yjkim721/STRIP-ViTA`

Table 7: The successful filtering rate (%) on the CIFAR-10 dataset.

| Method↓, Defense→ | Spectral Signatures | Activation Clustering |
|---|---|---|
| UBW-P | 10.96 | 52.61 |
| UBW-C | 9.40 | 20.51 |

Table 8: The resistance to MCR and NAD on the CIFAR-10 dataset.

| Defense→ | No Defense | | MCR | | NAD | |
|---|---|---|---|---|---|---|
| Method↓, Metric→ | BA (%) | ASR-A (%) | BA (%) | ASR-A (%) | BA (%) | ASR-A (%) |
| UBW-P | 90.59 | 92.30 | 88.17 | 96.20 | 67.98 | 99.40 |
| UBW-C | 86.99 | 89.80 | 86.15 | 79.10 | 77.13 | 36.00 |

Table 9: The effectiveness of our UBW-P with different types of triggers on the CIFAR-10 dataset.

| Method↓, Metric→ | BA (%) | ASR-A (%) | ASR-C (%) | $D_p$ |
|---|---|---|---|---|
| UBW-P (BadNets) | 90.59 | 92.30 | 92.51 | 2.2548 |
| UBW-P (WaNet) | 89.90 | 73.00 | 70.45 | 2.0368 |

codes with default settings on the CIFAR-10 dataset. Besides, we adopt the *successful filtering rate* defined as the number of filtered poisoned samples over that of all filtered samples as our evaluation metric. In general, the smaller the successful filtering rate, the more resistance of our UBW.

**Results.** As shown in Table 7, these defenses fail to filter our watermarked samples under both poisoned-label and clean-label to some extent. We speculate that it is mostly because poisoned samples generated by our UBW-P and UBW-C tend to scatter in the whole space instead of forming a single cluster in the feature space. We will further explore it in the future.

### G.5 Resistance to MCR and NAD

Here we discuss whether our methods are resistant to mode connectivity repairing (MCR) [28] and neural attention distillation (NAD) [29], which are two advanced repairing-based backdoor defenses.

**Settings.** We implement MCR and NAD based on the codes provided in `BackdoorBox`.

**Results.** As shown in Table 8, both our UBW-P and UBW-C are resistant to MCR and NAD to some extent. Their failures are probably because both of them contain a fine-tuning stage, which is ineffective for our UBWs (as demonstrated in Section 5.4.2).

## H   UBW-P with Imperceptible Trigger Patterns

In our main manuscript, we design our UBW-P based on BadNets-type triggers since it is the most straightforward method. We intend to show how simple it is to design UBW under the poisoned-label setting. Here we demonstrate that our UBW-P is still effective with imperceptible trigger patterns.

**Settings.** We adopt the advanced invisible targeted backdoor attack – WaNet [13] to design our UBW-P with imperceptible trigger patterns. We also implement it based on the open-sourced codes of vanilla WaNet provided in `BackdoorBox` [15]. Specifically, we set the warping kernel size as 16 and conduct experiments on the CIFAR-10 dataset. Except for the trigger patterns, all other settings are the same as those used in our standard UBW-P.

**Results.** As shown in Table 9, our UBW-P can still reach promising performance with imperceptible trigger patterns, although it may have relatively low ASR compared to UBW-P with the BadNets-type visible trigger. It seems that there is a trade-off between ASR and trigger visibility. We will discuss how to better balance the watermark effectiveness and its stealthiness in our future work.

## I   The Transferability of our UBW-C

Recall that in the optimization process of our UBW-C, we need to know the model structure $f$ in advance. Following the classical settings of bi-level-optimization-type backdoor attacks (*e.g.*, LIRA

Table 10: The performance of our UBW-C with different model structures trained on the watermarked CIFAR-10 dataset generated with ResNet-18.

| Metric↓, Model→ | ResNet-18 | ResNet-34 | VGG-16-BN | VGG-19-BN |
|---|---|---|---|---|
| BA (%) | 86.99 | 87.34 | 86.83 | 88.55 |
| ASR-A (%) | 87.56 | 78.89 | 75.80 | 74.30 |

[6] and Sleeper Agent [4]), we report the results of attacking DNN with the same model structure as the one used for generating poisoned samples. In practice, dataset users may adopt different model structures since dataset owners have no information about the model training. In this section, we evaluate whether the watermarked dataset is still effective in watermarking DNNs having different structures compared to the one used for dataset generation (*i.e.*, transferability).

**Settings.** We adopt ResNet-18 to generate a UBW-C training dataset, based on which to train different models (*i.e.*, ResNet-18, ResNet-34, VGG-16-BN, and VGG-19-BN). Except for the model structure, all other settings are the same as those used in Section 5.

**Results.** As shown in Table 10, our UBW-C has high transferability. Accordingly, our methods are practical in protecting open-sourced datasets.

## J  Connections and Differences with Related Works

In this section, we discuss the connections and differences between our UBW and adversarial attacks, data poisoning, and classical untargeted attacks. We also discuss the connections and differences between our UBW-based dataset ownership verification and model ownership verification.

### J.1  Connections and Differences with Adversarial Attacks

Both our UBW and adversarial attacks intend to make the DNNs misclassify samples during the inference process by adding malicious perturbations. However, they still have some intrinsic differences.

Firstly, the success of adversarial attacks is mostly due to the behavior differences between DNNs and humans, while that of our UBW results from the data-driven training paradigm and excessive learning ability of DNNs. Secondly, the malicious perturbations are known (*i.e.*, non-optimized) by UBW whereas adversarial attacks need to obtain them based on the optimization process. As such, adversarial attacks cannot to be real-time in many cases, since the optimization requires querying the DNNs multiple times under either white-box [30, 31, 32] or black-box [33, 34, 35] settings. Lastly, our UBW requires modifying the training samples without any additional requirements in the inference process, while adversarial attacks need to control the inference process to some extent.

### J.2  Connections and Differences with Data Poisoning

Currently, there are two types of data poisoning, including classical data poisoning [36, 37, 38] and advanced data poisoning [39, 40, 41]. Specifically, the former type of data poisoning intends to reduce model generalization, so that the attacked DNNs behave well on training samples whereas having limited effectiveness in predicting testing samples. The latter one requires that the model has good benign accuracy while misclassifying some adversary-specified unmodified samples.

Our UBW shares some similarities to data poisoning in the training process. Specifically, they all intend to embed distinctive prediction behaviors in the DNNs by poisoning some training samples. However, they also have many essential differences. The detailed differences are as follows:

**The Differences Compared with Classical Data Poisoning.** Firstly, UBW has a different goal compared with classical data poisoning. Specifically, UBW preserves the accuracy in predicting benign testing samples whereas classical data poisoning is not. Secondly, UBW is also more stealthy compared with classical data poisoning, since dataset users can easily detect classical data poisoning by evaluating model performance on a local verification set. In contrast, this method has limited benefits in detecting UBW. Lastly, the effectiveness of classical data poisoning is mostly due to the sensitiveness of the training process, so that even a small domain shift of training samples may lead to significantly different decision surfaces of attacked models. It is different from that of our UBW.

**The Differences Compared with Advanced Data Poisoning.** Firstly, advanced data poisoning can only misclassify a few selected images whereas UBW can lead to the misjudgments of all images containing the trigger pattern. It is mostly due to their second difference that data poisoning does not require modifying the (benign) images before the inference process. Thirdly, the effectiveness of advanced data poisoning is mainly because DNNs are over-parameterized, so that the decision surface can have sophisticated structures near the adversary-specified samples for misclassification. It is also different from that of our UBW.

## J.3   Connections and Differences with Classical Untargeted Attacks

Both our UBW and classical untargeted attacks (*e.g.*, untargeted adversarial attacks) intend to make the model misclassify specific sample(s). However, different from existing classical untargeted attacks which simply maximize the loss between the predictions of those samples and their ground-truth labels, our UBW also requires optimizing the prediction dispersibility so that the adversaries cannot deterministically manipulate model predictions. Maximizing only the untargeted loss may not be able to disperse model predictions, since targeted attacks can also maximize that loss when the target label is different from the ground-truth one of the sample. Besides, introducing prediction dispersibility may also increase the difficulty of the untargeted attack since it may contradict the untargeted loss to some extent (as described in Section B).

## J.4   Connections and Differences with Model Ownership Verification

Our UBW-based dataset ownership verification enjoys some similarities to model ownership verification [42, 43, 44] since they all conduct verification based on the distinctive behaviors of DNNs. However, they still have many fundamental differences, as follows:

Firstly, dataset ownership verification has different threat models and requires different capacities. Specifically, model ownership verification is adopted to protect the copyrights of open-sourced or deployed models, while our method is for protecting dataset copyrights. Accordingly, our UBW-based method only needs to modify the dataset, whereas model ownership verification usually also requires controlling other training components (*e.g.*, loss). In other words, our UBW-based method can also be exploited to protect model copyrights, whereas most of the existing methods for model ownership verification are not capable to protect (open-sourced) datasets.

Secondly, to the best of our knowledge, almost all existing black-box model ownership verification was designed based on the targeted attacks (*e.g.*, targeted poison-only backdoor attacks) and therefore introducing new security risks in DNNs. In contrast, our verification method is mostly harmless, since our UBW used for dataset watermarking is untargeted and with high prediction dispersibility.

## J.5   Connections and Differences with Radioactive Data

We notice that radioactive data (RD) [45] (under the black-box setting) can also be exploited as dataset watermarking for ownership verification by analyzing the loss of watermarked and benign images. If the loss of watermarked images is significantly lower than that of their benign version, RD treats the suspicious model as trained on the protected dataset. Both RD and UBW-C require knowing the model structure in advance, although they all have transferability. However, they still have many fundamental differences, as follows:

Firstly, our UBWs have a different verification mechanism compared to RD. Specifically, UBWs adopt the change of predicted probability on the ground-truth label, while RD exploits the loss change for verification. In practice, it is relatively difficult to select the confidence budget for RD since the loss values may change significantly across different datasets. In contrast, users can easily select the confidence budget (*i.e.*, $\tau$) from $[0, 1]$ since the predicted probability on the ground-truth label are relatively stable (*e.g.*, nearly 1 for benign samples).

Secondly, our UBWs require fewer defender capacities compared to RD. RD needs to have the prediction vectors or even the model source files for ownership verification, whereas UBWs only require the probability in the predicted label. Accordingly, our method can even be generalized to the scenario that users can only obtain the predicted labels (as suggested in [46]), based on examining whether poisoned images have different predictions compared to their benign version, whereas RD cannot. We will further discuss the label-only UBW verification in our future work.

Lastly, it seems that RD is far less effective on datasets with relatively low image resolution and fewer samples (*e.g.*, CIFAR-10)[4]. In contrast, our methods have promising performance on them.

## K    Discussions about Adopted Data

In this paper, all adopted samples are from the open-sourced datasets (*i.e.*, CIFAR-10 and ImageNet). The ImageNet dataset may contain a few personal contents, such as human faces. However, our research treats all objects the same and does not intentionally exploit or manipulate these contents. Accordingly, our work fulfills the requirements of those datasets and should not be regarded as a violation of personal privacy. Besides, our samples contain no offensive content, since we only add some invisible noises or non-semantic patches to a few benign images.