# OpenReview forum: "Untargeted Backdoor Watermark: Towards Harmless and Stealthy Dataset Copyright Protection"
_NeurIPS.cc/2022/Conference — NeurIPS 2022 Accept_

### Official Review · Reviewer_RZea · 2022-07-10

**Rating:** 5
**Confidence:** 3
**Ethics Flag:** Yes
**Soundness:** 3 good
**Presentation:** 2 fair
**Contribution:** 2 fair

**Summary:**

This paper proposes a methods, UBW-C and UBW-P, to verify unauthorized use of open-sourced dataset. For this, authors train a watermark generator G(. ; theta) using eq. 6 to minimize L(f(G(x:theta);w*),y) and entropy. Also, authors insist that the proposed watermarking is robust in dispersibility for predictions of poisoned data. A malicious network that trained using the watermarked dataset may predict randomly for watermarked test data and clearly for clean test data, so it is possible to verify using the difference between the two predictions, for watermarked and clean test data.

**Questions:**

I have some concerns as follow:

1) I cannot be sure about strength of the untargeted watermarking.  What is better than targeted one? In Fig 2, only difference between target label and random label in prediction.

2) Dispersibility occupies large portion of this paper, but I don't undestand necessity of it. It isn't used for training generator in Eq. 6 and 7, and used only as evaluation metric. However, I think it is entropy-based ASR instead of accuracy-based ASR, but they ar not much different.

3) Table 1 and 2 show that the proposed highly drops the benign accuracy from 92.53% to 86.99% for CIFAR10, and 67.3% to 59.6% for ImageNet subset. It seems that the proposed watermark is harmful.

4) Only ResNet-18 and small-scale datasets(CIFAR10 and ImageNet subset) were used in evaluation. Then, only label-consiststent watermarking and Sleeper Agent were compared. I think it is necessary to compare other recent works such as Radioactive Data [1] or which is much similar to the proposed method, or various backdoor attack/data poisoning approaches, and authors have to compare with them.

[1] Sablayrolles et. al.. Radioactive data: tracing through training. ICML 2020



5) In Eq. 6 and 7, f(.) means NN architecture. Is it necessary to know architecture of malicious model?





**Ethics Review Area:**

["Privacy and Security (e.g., consent)"]

**Limitations:**

1) Drop in BA for WBD-C

2) Unclear strength of random classification in dataset watermarking.

**Strengths And Weaknesses:**

This work has specialty in untargeted attack that makes malicious network predicts random label for watermarked test data. Then, experiment section provides comparisons with some other data poisoning methods on CIFAR10, and ImageNet subset. However, I think this paper is weak in comparisons to similar works under various conditions(architectures, and large-scaled datasets), and the proposed seems quite harmful for original dataset. Also, I think the proposed verification method requires detailed specification on malicious network, so it seems weak on practice.

===============After Author-Reviewer Discussion==========================
I raised some concerns at the first review, and many of them are discussed duing this period. In specific, issues about applications on large-scaled datasets, and transferability, stealthiness are well addressed.


Strengths: This work addresses an interesting which make a trained DNN predicts random label via dataset watermarking. I think the random prediction is an insightful and novel. Then, large-scaled datasets, and transferability, stealthiness, robustness to defense are well addressed.

Weaknesses: Drop in BA for WBD-C is a weakness, and I'm still not sure about whether the random classification is better than previous guided misclassification for verifying malicious users. I expect the property is fit to attacking purpose.


However, I think the insightful property is more important, and I'll expect the weakness can be enhanced later. Also, I'll re-update my score at the end of Reviewer-MetaReviewer Discussion.

---

> ### Author Response · Authors · 2022-08-02
> **Author Response (Part I)**
>
> We sincerely thank you for your valuable time and comments. We are encouraged by your positive comments on the specialty in our promising untargeted attack. We are deeply sorry for the misunderstandings that our paper may cause you. Please kindly find our clarifications below to your concerns.
>
> ---
> **Q1**: I cannot be sure about strength of the untargeted watermarking. What is better than targeted one? In Fig 2, only difference between target label and random label in prediction.
>
> **R1**: Thank you for this insightful question! Why we need the untargeted backdoor watermark is one of the core motivations of this paper and we are deeply sorry that we failed to make you fully understand it. The detailed strengths of our UBW are as follows:
> - In short, **our untargeted backdoor watermark is harmless compared to the existing targeted backdoor watermarks**. Existing targeted backdoor watermarks introduce new security threats in the model since the backdoor adversaries can determine model predictions of malicious samples. In contrast, the predictions generated by DNNs watermarked by our UBW are dispersible and therefore the adversaries cannot explicitly control model predictions.
> - **This harmlessness is necessary for both dataset owners and dataset users**. For the dataset users, they can use the watermarked dataset without fear of being attacked by the dataset owners who know how to activate the hidden backdoor; For the dataset owners, more users are willing to use their dataset. They are also excluded from suspicion when the models are attacked.
> - **Our methods can naturally bypass some backdoor detections** (*e.g.*, Neural Cleanse and Spectral Signatures) due to their untargeted nature. Accordingly, our methods are more stealthy, compared to targeted attacks.
>
> We are deeply sorry for the ambiguity that our paper may cause you again. We will add more details in both our introduction and the proposed method in the revision.
>
> ---
> **Q2**: Dispersibility occupies large portion of this paper, but I don't undestand necessity of it. It isn't used for training generator in Eq. 6 and 7, and used only as evaluation metric. However, I think it is entropy-based ASR instead of accuracy-based ASR, but they ar not much different.
>
> **R2**: Thank you for your question and we do understand your concerns. In our paper, we define three dispersibilities, including averaged prediction dispersibility ($D_p$), averaged sample-wise dispersibility ($D_s$), and averaged class-wise dispersibility ($D_c$). Why we need them is one of the core motivations of this paper and we are deeply sorry that we failed to make you fully understand it. Here we will further explain their necessities.
> - As we illustrated in the aforementioned R1, **the averaged prediction dispersibility ($D_p$) is necessary for harmless dataset watermarking**. This is why we include dispersibility as one of our watermark's goals in Section 3.2 (Line 146-156, Page 4) and treat it as one of the evaluation metrics.
> - However, as we explained in Section 3.4 (Line 177-179, Page5), **$D_p$ is non-differentiable and therefore cannot be optimized directly in UBW-C**. Accordingly, we introduce $D_s$ and $D_c$ as two differentiable surrogate dispersibilities to alleviate this problem.
> - Once we have $D_s$ and $D_c$, the remaining problem is how to design our UBW-C. According to our Theorem 1, **we can optimize the averaged sample-wise dispersibility $D_s$ and the class-wise dispersibility $D_c$ simultaneously by only maximizing $D_s$**. This is why we only include  $D_s$ in Eq. (6).
> - In Eq. (6), **the entropy is defined on the prediction vector of poisoned images** where their target and ground-truth labels are not involved. As such, it is not a simply (and trivial) entropy-based ASR as you thought.
>
> Given the aforementioned reasons, we can conclude that the design and theorem of dispersibilities are closely related to our methods and our methods are not the simple entropy-based ASR extensions. We are deeply sorry again for the ambiguity that our paper may cause you. We will add more details in the proposed method (Section 3.2-3.4) in the revision.
>
> ---

---

> > ### Author Response · Authors · 2022-08-02
> > **Author Response (Part II)**
> >
> > ---
> > **Q3**: Table 1 and 2 show that the proposed highly drops the benign accuracy from 92.53% to 86.99% for CIFAR10, and 67.3% to 59.6% for ImageNet subset. It seems that the proposed watermark is harmful.
> >
> > **R3**: Thank you for your comments and we do understand your concerns. We admit that our UBW-C has some decreases in benign accuracy, compared with the one trained on the benign dataset. This side effect is mostly due to the optimization process of bi-level optimization, which is relatively difficult in practice. We observed similar phenomena in methods which are also based on bi-level optimization (*e.g.*, Sleeper Agent and LIRA). We use the standard bi-level optimization with minimal modifications to make our methods and contributions more explicit. We believe that the decrease will not significantly reduce its practicality. Our next step is to explore how to better balance BA, ASR, and dispersibility. We hope that this acceptable BA gap will not eliminate our contributions and impacts as early work (with insights and theoretical supports) in this important field.
> >
> > ---
> > **Q4**: Only ResNet-18 and small-scale datasets(CIFAR10 and ImageNet subset) were used in evaluation.
> >
> > **R4**: Thank you for your comments and we do understand your concerns.
> > - We adopt ResNet-18 simply following the classical settings in backdoor-related methods. However, we do understand your concern about whether our methods are still effective with different model structures. To verify it, we evaluate our methods with VGG. As shown in Table 1, **our methods can still reach promising performance with different model structures**, although the performance may have some fluctuations.
> > - We adopt the ImageNet subset (with 50 classes) instead of the whole ImageNet due to the limitation of computational resources. However, we do understand your concern about whether our methods are still effective on large-scale datasets. To alleviate your concern, we train our UBW-P on the whole ImageNet dataset for only 30 epochs with the pre-trained model due to the limitation of time and computational resources. Since UBW-C takes more epochs due to the generation process of poisoned samples, we train it on the Tiny-ImageNet dataset (with 200 classes) due to the limitation of time and computational resources. As shown in Table 2, **our methods are still effective on large-scale datasets to some extent**. We notice that users can obtain better performance on the ImageNet dataset by training the model more epochs, especially training from the scratch.
> >
> > We will add more details and discussions in the appendix of our revision.
> >
> >
> > Table 1. The performance of our UBW with different model structures on CIFAR-10.
> > | Model$\downarrow$          | Attack$\downarrow$, Metric$\rightarrow$ |   BA   |  ASR-A |  ASR-C | $D_p$ |
> > |:---------:|:---------------------------------------:|:------:|:------:|:------:|:-----:|
> > | ResNet-18 |                  UBW-P                  | 90.59  | 92.30  | 92.51  | 2.2548  |
> > |   ResNet-18  |                  UBW-C                  | 86.99  | 89.80  | 87.56  | 1.2641  |
> > | VGG-16 |                  UBW-P                  |    91.25   |    88.20   |    86.46   |   2.0244   |
> > |   VGG-16  |                  UBW-C                  | 87.20  |    78.21   | 74.34  |   0.9875   |
> >
> >
> >
> > Table 2. The performance of our UBW on large-scale datasets.
> > | Attack$\downarrow$, Metric$\rightarrow$ |   BA   |  ASR-A |  ASR-C |  $D_p$  |
> > |:---------------------------------------:|:------:|:------:|:------:|:-------:|
> > |                  UBW-P                  |    71.36   |    50.00   |    42.56   |    1.8346    |
> > |                  UBW-C                  | 51.56  | 88.00  | 86.54  | 2.9871  |
> >
> >
> > ---

---

> > > ### Author Response · Authors · 2022-08-02
> > > **Author Response (Part III)**
> > >
> > > ---
> > > **Q5**: Only label-consistent watermarking and Sleeper Agent were compared. I think it is necessary to compare other recent works such as Radioactive Data which is similar to the proposed method, or various backdoor attack/data poisoning approaches.
> > >
> > > **R5**: Thank you for these comments and we do understand your concerns.
> > > - Firstly, we have to admit that we failed to find that Radioactive Data was also designed for dataset ownership verification. We sincerely thank you for pointing it out. After reading backdoor-embedding-based dataset watermarking (BEDW), radioactive data (RD), and papers that cited them, we can confirm that only BEWD and RD claimed that they can be used for dataset ownership verification. We will add RD to our related work in the revision.
> > > - Compared with RD, our UBW requires fewer user capacities and therefore is more practical. Specifically, RD is model-dependent since it requires users to have a fixed and known feature extractor for generating radioactive data. Besides, RD requires to have the prediction vectors or even the model source files for ownership verification, whereas we only need the probability in the predicted label. Accordingly, our method can even be generalized to the scenario that users can only obtain the predicted labels (by examining whether poisoned images have different predictions compared with their benign version) whereas RD cannot.
> > > - We mainly compared our UBW-C with label-consistent watermarking and Sleeper Agent, since they were the most representative and probably the only backdoor attacks under the clean-label setting. We have also compared with other backdoor attacks, including BadNets, Blended, and WaNet, under the poisoned-label setting.
> > > - However, we do understand your concerns that data poisoning may also be adapted for dataset watermarking since it can also introduce distinctive prediction behaviors. We have compared our methods with it in Section I of the appendix (Line 236-252, page 9-10). For example, the (advanced) data poisoning is also targeted and therefore is more harmful compared to our UBW.
> > >
> > >
> > > We will add more details and discussions in Section I (Connections and Differences with Related Works) of the appendix in the revision.
> > >
> > > ---
> > > **Q6**: In Eq. 6 and 7, f(.) means NN architecture. Is it necessary to know architecture of malicious model?
> > >
> > > **R6**: Thank you for this insightful question! Following the classical settings of bi-level-optimization-type backdoor attacks (*e.g.*, LIRA and Sleeper Agent), we report the results of attacking DNN with the same model architecture as the one used for generating poisoned samples. However, we do understand your concern about the transferability across different model architectures of our UBW-C. As shown in Table 3, our UBW-C has high transferability and therefore our method is practical in protecting released datasets.
> > >
> > > Table 3. The performance of our UBW-C with different model architectures trained on the poisoned dataset generated with ResNet-18.
> > > |       | ResNet-18 | ResNet-34 | VGG-16-BN | VGG-19-BN |
> > > |:-----:|:---------:|:---------:|:---------:|:---------:|
> > > |   BA  |   86.99   |   87.34   |   86.83   |   88.55   |
> > > | ASR-A |   87.56   |   78.89   |    75.80   |    74.30   |
> > >
> > > ---

---

> > > > ### Comment · Reviewer_RZea · 2022-08-05
> > > > **Post-Rebuttal Review**
> > > >
> > > > Thank you for your reply. It is much helpful to understand.
> > > >
> > > > But, I have some remained questions.
> > > >
> > > > 1)  For the untargeted attack's advantage, I agree that it is more stealthy; however, this benefit only applies if a malicious user can access to poisoned testset. If I were a dataset distributor, and I applied poisoning on my dataset, I would hide my poisoned test data. Therefore, malicious user cannot filter out by analyzing predictions of poisoned data, they can filter out only by analyzing input data.
> > > >
> > > > 2) Also, untargeted approach makes hard to verify watermarked DNN. For targeted approaches, they can verify by counting poisoned images that is classified as the target class. However, this approach have to count misclassified without any guided target class. I think this untargeted poisoning is not much different from Gaussian random noise injection that is small but can change predictions randomly. Therefore, I believe it is preferable for the dataset owner to adopt a controllable method.
> > > > Due to this difficuly to verify, a new metric, Dispersibility, is proposed, but I think that authors should suggest a practical advantages of random classification for poisoned images comparing targeted classification.
> > > >
> > > >
> > > > 3) I'm still unsure about harmlessness. However, Table 1 and 2 show that the proposed achieves higher ASRs, but don't show the harmlessness.
> > > > At table 1 and 2, UBW reports similar BA to targeted approaches'. Also, Table 2 in reply shows almost 20% drop in large-scale dataset, and 20% drop is not neglectable.
> > > >
> > > >
> > > >
> > > > 4) For Sleeper Agent, table 1 reports much different performances comparing their original paper. As described Table 2 and 3 of Sleeper Agent original paper, it achieved much better BA/ASR for CIFAR10 and ResNet18(the same condition to this paper).  The only difference is ratio of budget for poisoning.

---

> > > > > ### Author Response · Authors · 2022-08-05
> > > > > **Thank you for your feedback and our further explanations (Part I)**
> > > > >
> > > > > We greatly appreciate your feedback on our rebuttal and the further insightful questions and comments. Please kindly find our explanations about the remaining concerns as follows:
> > > > >
> > > > > ---
> > > > > **Q1**: For the untargeted attack's advantage, I agree that it is more stealthy; however, this benefit only applies if a malicious user can access to poisoned testset. If I were a dataset distributor, and I applied poisoning on my dataset, I would hide my poisoned test data. Therefore, malicious user cannot filter out by analyzing predictions of poisoned data, they can filter out only by analyzing input data.
> > > > >
> > > > > **R1**: Thank you for this comment and we do understand your concerns. However, we are deeply sorry for the misunderstandings that our response may cause you. In our previous R1, we argued that our methods can naturally bypass some backdoor detections (e.g., Neural Cleanse and Spectral Signatures) due to their untargeted nature and therefore are more stealthy. We notice that both **Neural Cleanse and Spectral Signatures are detection methods used for filtering poisoned training samples instead of poisoned testing samples**. Accordingly, this benefit of stealiness does not require the dataset owner to release their poisoned testset.
> > > > >
> > > > > Sorry again for the misunderstandings that our rebuttal may cause you and we will add more details and discussions in the appendix of our revision.
> > > > >
> > > > >
> > > > > ---
> > > > > **Q2**: Untargeted approach makes hard to verify watermarked DNN. I think this untargeted poisoning is not much different from Gaussian random noise injection which is small but can change predictions randomly. Therefore, I believe it is preferable for the dataset owner to adopt a controllable method.
> > > > >
> > > > > **R2**: Thank you for these comments and we are deeply sorry for the misunderstandings that our paper or response may cause you.
> > > > > - Firstly, we respectfully disagree that our work is not much different from Gaussian random noise injection which is small but can change predictions randomly. **Adding small random Gaussian noises to benign images will not significantly change model predictions**. Instead, using UBW triggers can activate the hidden backdoors of watermarked models and therefore change predictions randomly. In addition, **the trigger patterns used for backdoor activation are pre-defined and therefore more controllable** (compared to random noises).
> > > > > - Secondly, as we mentioned in R1 of our previous rebuttal,  although they are easier for watermarking and verification, **targeted attacks will introduce new threatening security risks** since the adversaries can determine the predictions of malicious samples.
> > > > >
> > > > > Accordingly, the untargeted backdoor watermarking is practical.
> > > > >
> > > > >
> > > > > ---
> > > > > **Q3**: Untargeted approach makes hard to verify watermarked DNN. Due to the difficulty for verification, a new metric, Dispersibility, is proposed, but I think that authors should suggest practical advantages of random classification for poisoned images comparing targeted classification.
> > > > >
> > > > > **R3**: Thank you for this constructive suggestion!
> > > > > - Our UBWs are harmless since the abnormal model behaviors are not deterministic. As such, they are more likely to be used in practice.
> > > > > - Our UBWs are more stealthy since they can naturally bypass many backdoor detection methods. Accordingly, the malicious dataset users can hardly notice our watermark helping us to keep the watermark in trained models.
> > > > >
> > > > >
> > > > > ---

---

> > > > > > ### Comment · Reviewer_RZea · 2022-08-06
> > > > > > **Post-Rebuttal Review2**
> > > > > >
> > > > > > Thanks for your reply, and some of my concerns are addressed.
> > > > > >
> > > > > > Basically, I think it is more important to compare with clean-labeled attacks. Actually, the poison-label attacks can be detected by human eye without defence methods. In my opinion, clean-label is necessary to be used in practice,
> > > > > >
> > > > > > 1) For BA drops, I agree on that the UBW-P is not much harmful, but I'm still have concerns about UBW-C. It drops about 6% of accuracy on CIFAR10 experiments. Then, authors mentioned that they reimplemented Sleeper Agent for fair comparison. According to Appendix, change of poisoning ratio, sample-specific trigger and alternative optimization are stated. However, I'm confused about the sample-specific trigger. As I know, data poisoning approaches doesn't use many triggers, I expect that only a trigger per class is needed. I'm sorry for repeat of this question even authors provided results on TinyImageNet, but results on TinyImageNet is not included in Sleeper Agent and Label-consistent. Then, about 92% BA is reported in Sleeper Agent on CIFAR10 and ResNet18, but this paper reports 86.99% BA. For CIFAR10, this 5% is not small. Could you explain the modification of reimplementation in detail?
> > > > > >
> > > > > > 2) In Appendix F, authors only compared UBW-C and UBW-P with only BadNets using easy trigger, and I accept that the proposed is better than BadNets thanks to untargeted property. As I mentioned, I think the clean-label is necessary, so I focused on clean-labeled approaches. Sleeper Agent paper also provided results after defence methods including neural cleanser.  According to Sleeper Agent, the Neural Cleanser is not so good for detection of any of the backdoored classes. Therefore, I think comparison of UBW-C and clean-labeled approaches in image-level similarity, and results of defence.
> > > > > >
> > > > > > 3) For my mention about Gaussian Noise, it was not for devaluation or blame. If you felt uncomfortable, I'm sorry for that mention. I meant that this random prediction works similarly to Gaussian Noiseinjection, and I have a question about evidence capacity comparing targeted prediction. If I were a malicious user, I would insist that this misclassification is due to noise, and Gaussian Noise can lead random misclassification. Of course, I agree on your reply "Adding small random Gaussian noises to benign images will not significantly change model predictions.", but malicious user may insist as that. And, a judge cannot conclude this is unauthorized use of the watermarked dataset because of the random noise case. So, I'm worry about clear distinguish of misclassifications by this watermark, or random noise. For  targeted prediction case, the predictions are guided toward dataset owner's pre-defined class, so it is better in evidence ability as I thought.
> > > > > >
> > > > > >
> > > > > > 4) I'm not clearly understand about this comment "Our UBWs are harmless since the abnormal model behaviors are not deterministic." Any malicious user cannot have watermarked testset anyway, so I think they cannot distinguise whether their model works deterministically or randomly.
> > > > > >
> > > > > >
> > > > > > 5) Even though I thought as 3) and 4) , I think this approach is very interesting. As I thought, this random prediction by watermark is proper to password on treat model rather than dataset watermarking. If a treat model trained by the watermarked dataset works well on only image with trigger, and works poorly on clean image, it would be very useful.

---

> > > > > > > ### Author Response · Authors · 2022-08-06
> > > > > > > **Response to Post-Rebuttal Review 2 (Part I)**
> > > > > > >
> > > > > > > We greatly appreciate your positive feedback on our rebuttal and further insightful questions and comments. We are encouraged that you finally recognize our UBW methods. Please kindly find our explanations about your remaining concerns as follows:
> > > > > > >
> > > > > > > ---
> > > > > > > **Q1**: For BA drops, I agree on that the UBW-P is not much harmful, but I'm still have concerns about UBW-C. It drops about 6% of accuracy on CIFAR10 experiments.
> > > > > > >
> > > > > > > **R1**: Thank you for your comment and we do understand your concerns. We need to notice that **there is a trade-off between the BA and the ASR to some extent**, due to the poisoning rate $\gamma$ that users may adopt. As we illustrated in Figure 4 in our main manuscript, the BA decreases while the ASR increases with the increase of $\gamma$. We adopt $\gamma=10\%$ simply following the classical setting used in backdoor-related works. The BA drop is no more than 3% while the ASR is higher than 50% for both UBW-P and UBW-C if we set $\gamma=2\%$ on the CIFAR-10 dataset. We will further explore how to better balance the BA and ASR in our future work.
> > > > > > >
> > > > > > > ---
> > > > > > > **Q2**: According to Appendix, change of poisoning ratio, sample-specific trigger, and alternative optimization are stated. However, I'm confused about the sample-specific trigger. As I know, data poisoning approaches doesn't use many triggers, I expect that only a trigger per class is needed.
> > > > > > >
> > > > > > > **R2**: Thank you for this insightful question! In general, **using sample-specific trigger patterns are more effective and stealthy**, compared to using sample-agnostic one. Specifically,
> > > > > > > - **Effectiveness**: From the perspective of optimization, using sample-specific triggers introduces more variables for optimization and therefore makes the attack more effective. This benefit is especially critical for clean-label backdoor attacks since they are significantly more difficult compared to poisoned-label backdoor attacks. Specifically, the 'robust features' related to the target label will hinder the learning of backdoor triggers. This is the main reason why Sleeper Agent is more effective than label-consistent backdoor attack.
> > > > > > > - **Stealthiness**: As illustrated in [1, 2], most of backdoor defenses (e.g., Neural Cleanse) were designed based on a latent assumption that the trigger pattern is sample-agnostic. Accordingly, attacks with sample-specific triggers can easily bypass them since they break their fundamental assumption.
> > > > > > >
> > > > > > >
> > > > > > > Besides, we need to notice that the perturbations of modified images in advanced data poisoning (e.g., [3]) are also sample-specific. We will add more details and discussions in the appendix of our revision.
> > > > > > >
> > > > > > >
> > > > > > > References
> > > > > > > 1. Invisible Backdoor Attack with Sample-Specific Triggers. ICCV, 2021.
> > > > > > > 2. Input-Aware Dynamic Backdoor Attack. NeurIPS, 2020.
> > > > > > > 3. Witches' Brew: Industrial Scale Data Poisoning via Gradient Matching. ICLR, 2021.
> > > > > > >
> > > > > > >
> > > > > > > ---
> > > > > > > **Q3**: I'm sorry for repeat of this question even authors provided results on TinyImageNet, but results on TinyImageNet is not included in Sleeper Agent and Label-consistent. Then, about 92% BA is reported in Sleeper Agent on CIFAR10 and ResNet18, but this paper reports 86.99% BA. For CIFAR10, this 5% is not small. Could you explain the modification of reimplementation in detail?
> > > > > > >
> > > > > > > **R3**: Thank you for your question and we do understand your concern. Firstly, we are deeply sorry for the misunderstandings that our response may cause you. We did not intend to modify the implementation of Sleeper Agent. The point we were trying to make during our previous rebuttal was that we may have some differences in the implementation, even though we reproduced it based on its official codes and original paper. Besides, as we mentioned in our last rebuttal, we use a different trigger pattern, which also leads to different results.
> > > > > > >
> > > > > > > ---

---

> > > > > > > ### Author Response · Authors · 2022-08-06
> > > > > > > **Response to Post-Rebuttal Review 2 (Part II)**
> > > > > > >
> > > > > > > ---
> > > > > > > **Q4**: In Appendix G, the authors only compared UBW-C and UBW-P with only BadNets using easy trigger, and I accept that the proposed is better than BadNets thanks to untargeted property. As I mentioned, I think clean-label is necessary, so I focused on clean-labeled approaches. Sleeper Agent paper also provided results after defense methods including neural cleanser. According to Sleeper Agent, the Neural Cleanser is not so good for the detection of any of the backdoored classes. Therefore, I think the comparison of UBW-C and clean-labeled approaches in image-level similarity and results of defense are necessary.
> > > > > > >
> > > > > > >
> > > > > > > **R4**: Thank you for this constructive suggestion! We compared our methods with BadNets simply because it can be detected by almost all defenses. Accordingly, we can use its results for reference to better illustrate why our watermarks are resistant to the discussed defenses. However, we do understand your concerns and conduct additional experiments to verify whether Sleeper Agent and label-consistent backdoor attack are also resistant to potential defense methods. The results are summarized as follows:
> > > > > > >
> > > > > > > - The Resistance to Trigger Synthesis based Defenses: As you mentioned, **Sleeper Agent is also resistant to trigger synthesis based defenses, whereas label-consistent attack is not.** This is mostly because the trigger patterns used in the training process of Sleeper Agent are sample-specific, whereas that of label-consistent attack is sample-agnostic.
> > > > > > > - The Resistance to Saliency-based Defenses: **Both Sleeper Agent and label-consistent backdoor attacks can be detected by saliency-based defenses** since their trigger patterns used in the inference process are both sample-agnostic and both of them are targeted. Note that the trigger pattern adopted for Sleeper Agent in the inference process is sample-agnostic, although those used in the training process are sample-specific.
> > > > > > > - The Resistance to STRIP: **Both Sleeper Agent and label-consistent backdoor attacks can be detected by saliency-based defenses** since their trigger patterns used in the inference process are both sample-agnostic and both of them are targeted. Note that Sleeper Agent may resistant to STRIP to some extent, if random position mode is adopted.
> > > > > > >
> > > > > > >
> > > > > > >
> > > > > > > We will add more details and discussions in Appendix G in our revision.
> > > > > > >
> > > > > > > ---
> > > > > > > **Q5**: For my mention about Gaussian Noise, it was not for devaluation or blame. If you felt uncomfortable, I'm sorry for that mention. I meant that this random prediction works similarly to Gaussian Noiseinjection, and I have a question about evidence capacity comparing targeted prediction. If I were a malicious user, I would insist that this misclassification is due to noise, and Gaussian Noise can lead to random misclassification. Of course, I agree with your reply "Adding small random Gaussian noises to benign images will not significantly change model predictions.", but the malicious user may insist on that. And, a judge cannot conclude this is unauthorized use of the watermarked dataset because of the random noise case. So, I'm worried about a clear distinguish of misclassifications by this watermark, or random noise. For targeted prediction case, the predictions are guided toward dataset owner's pre-defined class, so it is better in evidence ability as I thought.
> > > > > > >
> > > > > > > **R5**: Thank you for your detailed explanations and insightful comments! We fully understand your concerns now. We agree that targeted watermarks  are easier for distinction in the verification stage. We argue that untargeted watermarks are also distinctive (and therefore are also practical), as follows:
> > > > > > > - We use one trigger pattern (e.g., white-black square) instead of different (random) noises to generate watermarked testing samples for verification. **It is unlikely that using a specific trigger pattern can shift the predictions of many different testing images if the suspicious model is not watermarked**.
> > > > > > > - We notice that the dataset owners have the benign version of their released watermarked dataset. Accordingly, **dataset owners can train DNNs on the benign dataset and show that the trigger pattern cannot change their predictions to refute the insistence of malicious users**.
> > > > > > >
> > > > > > > Thank you again for this insightful question. We will add more discussions in the appendix of our revision.
> > > > > > >
> > > > > > > ---

---

> > > > > > > ### Author Response · Authors · 2022-08-06
> > > > > > > **Response to Post-Rebuttal Review 2 (Part III)**
> > > > > > >
> > > > > > >
> > > > > > > ---
> > > > > > > **Q6**: I'm not clearly understand about this comment "Our UBWs are harmless since the abnormal model behaviors are not deterministic." Any malicious user cannot have watermarked testset anyway, so I think they cannot distinguise whether their model works deterministically or randomly.
> > > > > > >
> > > > > > > **R6**: Thank you for this question and we do understand your concern. Firstly, we are deeply sorry for the misunderstandings that this sentence may cause you. **The latent adversary in this sentence is not the malicious dataset users you thought**. Instead, we wanted to indicate that the **dataset owners may attack DNNs trained on their released dataset (with their pre-defined watermarks) if they are malicious**. In this case, the targeted backdoor watermarks may be very harmful since abnormal model behaviors are deterministic. We will add more details and explanations in the introduction and the proposed method in our revision.
> > > > > > >
> > > > > > > ---
> > > > > > > **Q7**: Even though I thought as Q5-6, I think this approach is very interesting. As I thought, this random prediction by watermark is proper to password on treat model rather than dataset watermarking. If a treat model trained by the watermarked dataset works well on only image with trigger, and works poorly on clean image, it would be very useful.
> > > > > > >
> > > > > > >
> > > > > > > **R7**: Thank you for your recognition and interesting perspective! Using a backdoor watermark as the password to protect DNNs is a promising research topic. Intuitively, it can be treated as the dual of our discussed problem since we have reverse goals regarding the inference process. It seems that we can also formulate this problem as a bi-level optimization and solve it with similar techniques adopted in this paper. However, it is out of the scope of this paper. We will discuss it in our future work.
> > > > > > >
> > > > > > >
> > > > > > > ---
> > > > > > > **Note**: We believe that your current negative score is mostly due to the misunderstands that our previous paper version may cause you. We sincerely thank you for your valuable time and comments, which greatly help us for improving our work. However, we think we may have addressed (most of) your main concerns. We would be very grateful if you can kindly update your score based on our clarifications and discussions. We are also happy to address your further questions and concerns before the rebuttal ends.
> > > > > > >
> > > > > > > ---

---

> > > > > > > ### Author Response · Authors · 2022-08-08
> > > > > > > **Thank you for your additional feedback**
> > > > > > >
> > > > > > > Please accept our appreciation for your positive feedback on our rebuttal and further insightful questions and comments. We are encouraged that you finally recognize our UBW methods.
> > > > > > >
> > > > > > > We believe that your current negative score is mostly due to the misunderstands that our previous paper version may cause you. We sincerely thank you for your valuable time and comments, which greatly help us for improving our work. However, we think we may have addressed (most of) your main concerns. We would be very grateful if you can kindly update your score based on our clarifications and discussions. We are also happy to address your further questions and concerns before the rebuttal ends.

---

> > > > > > > > ### Comment · Reviewer_RZea · 2022-08-08
> > > > > > > > **Thank you for your kind and patient replies**
> > > > > > > >
> > > > > > > > Thank you for your kind and patient replies, and it was very enjoyable discussion. Most of my concerns are addressed, and I know that I had some misunderstand. But, it is hard to definitively mention about my final score now because of a remained review preocedure. As I know, we will have Reviewer-MetaReviewer discussion period until tomorrow, and I will fix score considering your replies, and discussion during the period (unless this is my misconstruction about review procedure...).
> > > > > > > >
> > > > > > > > I'm sorry I can't give you a definite answer.

---

> > > > > > > > > ### Author Response · Authors · 2022-08-08
> > > > > > > > > **Thank you for your recognition and some kind explanations of the reviewing procedures.**
> > > > > > > > >
> > > > > > > > > Thank you for your recognition of our discussions and kind explanations. We do respect your decision and are willing to wait for your final score after the Reviewer-Metareviewer discussion ends. However, just for a warm notification, we think you may have some misunderstandings about the reviewing procedures. (PS: We have served as the reviewer of NeurIPS and ICLR on OpenReview many times and joined Reviewer-Metareviewer discussions multiple times.)
> > > > > > > > >
> > > > > > > > > In general, reviewers will change their pre-rebuttal score before the author-reviewer discussion period ends, if they think the authors have addressed their main concerns. In particular, this score is not the final score you thought, since reviewers can still change their scores during the Reviewer-Metareviewer discussion. This updated score is a reflection of your attitude toward the paper after the author-reviewer discussion. Otherwise, other reviewers cannot fully know what you think and therefore may not have an effective Reviewer-Metareviewer discussion.
> > > > > > > > >
> > > > > > > > > However, as we mentioned before, we sincerely thank you for helping us to improve our work and totally respect your decision of updating the score later :)

---

> > > > > ### Author Response · Authors · 2022-08-05
> > > > > **Thank you for your feedback and our further explanations (Part I)**
> > > > >
> > > > >
> > > > > ---
> > > > > **Q4**: Table 1 and 2 show that the proposed achieves higher ASRs, but don't show the harmlessness. At table 1 and 2, UBW reports similar BA to targeted approaches'. Also, Table 2 in reply shows almost 20% drop in large-scale dataset, and 20% drop is not neglectable.
> > > > >
> > > > > **R4**: Thank you for your questions and we believe there are some misunderstandings here.
> > > > > - Table 1-2 provided in our previous response are used to show that our methods are still effective under different model structures and on large-scale datasets. We did not intend to verify that our methods are harmless by showing these tables.
> > > > > - We are deeply sorry for the misunderstandings that our response may cause you. Specifically, **our Table 2 in the previous rebuttal may mislead you to think that our UBW-C causes approximately 20% BA drop on the large-scale dataset, compared with our UBW-P**. However, as we mentioned in the previous R5, **the UBW-P was trained on the whole ImageNet while UBW-C was trained on Tiny-ImageNet** (due to the limitation of time and computational resources). Accordingly, comparing their results are meaningless. To verify that our methods (UBW-P and UBW-C) will not significantly reduce the BA of the watermarked models, we provide additional results as follows:
> > > > >
> > > > > Table 1. The performance of our UBW-P on the whole ImageNet dataset.
> > > > > | Attack$\downarrow$, Metric$\rightarrow$ |   BA   |  ASR-A |  ASR-C |  $D_p$  |
> > > > > |:---------------------------------------:|:------:|:------:|:------:|:-------:|
> > > > > |                  No Attack                  | 72.29  | NA  | NA  | NA  |
> > > > > |                  UBW-P                  |    71.36   |    50.00   |    42.56   |    1.8346    |
> > > > >
> > > > >
> > > > > Table 2. The performance of our UBW-C on the Tiny-ImageNet dataset.
> > > > > | Attack$\downarrow$, Metric$\rightarrow$ |   BA   |  ASR-A |  ASR-C |  $D_p$  |
> > > > > |:---------------------------------------:|:------:|:------:|:------:|:-------:|
> > > > > |                  No Attack                  | 54.04  | NA  | NA  | NA  |
> > > > > |                  UBW-C                  | 51.56  | 88.00  | 86.54  | 2.9871  |
> > > > >
> > > > >
> > > > >
> > > > > Sorry again for the misunderstandings that our rebuttal may cause you and we will add more details and discussions in the appendix of our revision.
> > > > >
> > > > > ---
> > > > > **Q5**: For Sleeper Agent, table 1 reports much different performances comparing their original paper. As described Table 2 and 3 of Sleeper Agent original paper, it achieved much better BA/ASR for CIFAR10 and ResNet18 (the same condition to this paper). The only difference is ratio of budget for poisoning.
> > > > >
> > > > >
> > > > >
> > > > > **R5**: Thank you for this question and we do understand your concerns about whether the comparisons are fair. As we illustrated in Appendix, the optimization process of our UBW-C has some similarities to that of the Sleeper Agent. Accordingly, **we modified the codes of Sleeper Agent to implement our UBW-C for ensuring a fair comparison**. However, we found that the codes of Sleeper Agent are too structive for modification and therefore we have to re-implement its codes. The re-implementation process may introduce some differences and therefore cause different results. Besides, **we used a different trigger pattern**, compared to the one used in the original paper of Sleeper Agent. It also leads to different results.
> > > > >
> > > > > ---

---

### Official Review · Reviewer_LiAP · 2022-07-10

**Rating:** 7
**Confidence:** 4
**Soundness:** 3 good
**Presentation:** 3 good
**Contribution:** 3 good

**Summary:**

The paper aimed to protect open-source datasets from illegal DNN training by injecting verifiable backdoor watermarks. It first revealed that the existing backdoor watermarking techniques used targeted labels and could be exploited by adversaries for attacks. It then proposed novel untargeted backdoor watermarking techniques that are both effective and harmless in poisoned-label (UBW-P) and clean-label (UBW-C) settings. UBW-P stamped simple backdoor triggers on a subset of data and changed their labels randomly. UBW-C did not modify image labels; it instead optimized the trigger generation function in a bi-level optimization. The proposed methods were verified on CIFAR-10 and ImageNet, showing high effectiveness and dispersibility while being stealthy to standard backdoor defenses, including Fine-tuning, Fine-pruning, NeuralCleanse, STRIP, and GradCam inspection.

**Questions:**

- Can we design imperceptible triggers for UBW-P and UBW-C?
- Are UBW-P and UBW-C stealthy under dataset-based backdoor defenses such as Spectral Signatures and Activation Clustering?
- Are UBW-P and UBW-C stealthy under recent backdoor defenses such as NAD and Mode Connectivity?

**Limitations:**

The authors discusses the societal impacts of the proposed technique. I would recommend to add limitations mentioned above.

**Strengths And Weaknesses:**

## Strengths:
- The authors provided a good discussion on the issue of the common targeted backdoor watermarks and why we needed untargeted backdoor watermarks.
- The proposed untargeted backdoor watermarking techniques are novel and useful. I believe the topic of untargeted backdoors is easy to think of. However, it is not attractive and unexplored in the context of backdoor attacks due to its unpredictability. However, the authors found data watermarking a suitable application for these techniques, in which unpredictability becomes a strength when considering safety.
- The paper designed two versions for both poisoned-label and clean-label settings.
- The authors proposed new dispersibility metrics, which are technically sound. They are particularly helpful in designing the clean-label untargeted backdoor watermarks (UBW-C).
-   The proposed methods were verified on CIFAR-10 and ImageNet, showing high effectiveness and dispersibility while being stealthy to standard backdoor defenses, including Fine-tuning, Fine-pruning, NeuralCleanse, STRIP, and GradCam inspection.


## Weaknesses:
- The proposed watermarks are perceptible by human subjects and can be manually removed. The authors should try the imperceptible techniques:
   - In UBW-P, instead of using the BadNets triggers, the authors can try the imperceptible ones from WaNet, or LIRA [1].
   - In UBW-C, there is no constraint to enforce the poisoned image to be similar to the clean one. As a result, we can see apparent artifacts on the poisoned examples. Employing that constraint in the optimization goal (Eq. 6) would be an interesting direction to explore.
- The authors should verify the proposed watermarking methods under dataset-based backdoor defenses such as Spectral Signatures [2] and Activation Clustering [3].
- It is also more persuasive if they are verified under more backdoor defenses such as NAD [4] and Mode Connectivity [5].
- The denotations for the datasets (\mathcal{D}) and dispersibility metrics (D) are easy to be confused.

[1] Lira: Learnable, imperceptible and robust backdoor attacks. In CVPR 2021.

[2] Spectral signatures in backdoor attacks. In NeurIPS 2018.

[3] Detecting Backdoor Attacks on Deep Neural Networks by Activation Clustering. In SafeAI at AAAI 2019

[4] Neural attention distillation: Erasing backdoor triggers from deep neural networks. In ICLR 2021.

[5] Bridging mode connectivity in loss landscapes and adversarial robustness. In ICLR 2020.

---

> ### Author Response · Authors · 2022-08-02
> **Author Response (Part I)**
>
> We sincerely thank you for your valuable time and comments. We are encouraged by your positive comments on our **good motivation**, **novelty**, **well-designed methods and metrics**, **effectiveness**, and **resistance to standard backdoor defenses**. We will alleviate your remaining concerns as follows:
>
>
> ---
> **Q1**: The proposed watermarks are perceptible by human subjects and can be manually removed. In UBW-P, instead of using the BadNets triggers, the authors can try the imperceptible ones from WaNet or LIRA.
>
> **R1**: Thank you for this insightful comment and constructive suggestion! We designed our UBW-P based on BadNets-type triggers simply because it is the most straightforward method. We intended to show how simple it is to design UBW under the poison-label setting. However, we do understand your concern and agree that imperceptible UBW-P would be better for its stealthiness. Following your suggestions, we evaluate the UBW-P with WaNet-type triggers. As shown in Table 1, **our UBW-P can still reach promissing performance with imperceptible trigger patterns**. We will add more details and discussions in the appendix of our revision.
>
>
> Table 1. The effectiveness of our UBW-P with different types of triggers on CIFAR-10.
> | Method$\downarrow$, Metric$\rightarrow$ | BA (\%) | ASR-A (%) | ASR-C (\%) |  $D_p$ |
> |:---------------------------------------:|:-------:|:---------:|:----------:|:------:|
> |             UBW-P (BadNets)             |  90.59  |   92.30   |   92.51    | 2.2548 |
> |              UBW-P (WaNet)              |    89.90    |     73.00     |      70.45     |    2.0368   |
>
>
>
>
> ---
> **Q2**: The proposed watermarks are perceptible by human subjects and can be manually removed. In UBW-C, there is no constraint to enforce the poisoned image to be similar to the clean one. As a result, we can see apparent artifacts on the poisoned examples. Employing that constraint in the optimization goal (Eq. 6) would be an interesting direction to explore.
>
> **R2**: Thank you for this insightful comment and constructive suggestion! Firstly, we are deeply sorry for the misunderstanding that our paper caused you. In fact, **we did ensure that the poisoned image $x'$ should be similar to its benign version $x$** by requiring $||x'-x||_\infty < \epsilon$ where $\epsilon$ is the perturbation budget. We included these details in Appendix Section B (Page 2, Line 25-26) due to the space limitation of our main manuscript. As shown in the following Table 2, as we expected, using a larger $\epsilon$ can increase ASR (with the sacrifice of some stealthiness degrees). We will add more details in Section 3.4 of our main manuscript and the appendix in the revision.
>
>
> Table 2. The effectiveness of our UBW-C with different perturbation budgets on CIFAR-10.
> | Method$\downarrow$, Metric$\rightarrow$ | BA (\%) | ASR-A (%) | ASR-C (\%) |  $D_p$ |
> |:---------------------------------------:|:-------:|:---------:|:----------:|:------:|
> |              UBW-C (16/255)             |  86.99  |   89.80   |   87.56    | 1.2641 |
> |              UBW-C (32/255)             |    86.17    |     94.55     |      92.03     |    1.0112   |
>
>
>
> Besides, we also notice the artifact that you pointed out. We think **it is most probably due to the differences between the $\ell^\infty$ norm and the human visual system** since we observed similar problems in existing attacks with $\ell^\infty$-bounded additive perturbations. We will explore how to further enhance the stealthiness of UBW-C in our future work.
>
>
> ---

---

> > ### Author Response · Authors · 2022-08-02
> > **Author Response (Part II)**
> >
> >
> > ---
> > **Q3**: The authors should verify the proposed watermarking methods under dataset-based backdoor defenses such as Spectral Signatures and Activation Clustering.
> >
> > **R3**: Thank you for this constructive suggestion! Both spectral signatures and activation clustering tend to filter poisoned samples from the training dataset, based on sample behaviors in hidden feature space. These methods rely on a latent assumption that poisoned samples will form a separate cluster in the hidden feature space. This assumption usually holds in existing targeted poison-only backdoor attacks. However, as we illustrated in Section H (Figure 5) in the appendix, **poisoned samples generated by our untargeted UBW-P and UBW-C tend to scatter in the whole space instead of forming a single cluster. Accordingly, our methods are naturally resistant to both spectral signatures and activation clustering**. To verify it, we conduct some experiments on the CIFAR-10 dataset. As shown in the following Table 3, these defenses fail to filter our poisoned samples to some extent. We will add more details and discussions in the appendix of our revision.
> >
> >
> > Table 3. The successful filtering rate (the number of filtered poisoned samples / the number of all filtered samples, %) on CIFAR-10.
> >
> > | Attack$\downarrow$, Defense$\rightarrow$ | SS | AC |
> > |:----------------------------------------:|:--:|:--:|
> > |                   UBW-P                  |  10.96% (548/5000) |  52.61% (4981/9467) |
> > |                   UBW-C                  |  9.40% (470/5000) |  20.51% (1003/4889) |
> >
> >
> >
> > ---
> > **Q4**: It is also more persuasive if they are verified under more backdoor defenses such as Neural Attention Distillation (NAD) and Mode Connectivity Repairing (MCR).
> >
> > **R4**: Thank you for this constructive suggestion! We evaluate our methods on the CIFAR-10 dataset. As shown in Table 4-5, **both UBW-P and UBW-C are resistant to NAD and MCR to some extent**. Their failures are probably because both NAD and MCR contain a fine-tuning stage, which is ineffective for our UBW. We will add more details and discussions in the appendix of our revision.
> >
> > Table 4. The resistance to NAD on CIFAR-10.
> > | Attack$\downarrow$, Metric$\rightarrow$ | BA | ASR-A | ASR-C |
> > |:---------------------------------------:|:--:|:-----:|:-----:|
> > |                  UBW-P                  |  67.98 |   99.40   |   89.87   |
> > |                  UBW-C                  |  77.13 |   36.00   |   29.81   |
> >
> >
> >
> > Table 5. The resistance to MCR on CIFAR-10.
> > | Attack$\downarrow$, Metric$\rightarrow$ | BA | ASR-A | ASR-C |
> > |:---------------------------------------:|:--:|:-----:|:-----:|
> > |                  UBW-P                  |  88.17 |   96.20   |   96.06   |
> > |                  UBW-C                  |  86.15 |   79.10   |   71.69   |
> >
> >
> > ---
> > **Q5**: The denotations for the datasets $\mathcal{D}$ and dispersibility metrics $D$ are easy to be confused.
> >
> > **R5**: Thank you for pointing it out! We will change the dispersibility metrics from $D$ to $d$ in our revision.
> >
> > ---

---

> ### Author Response · Authors · 2022-08-04
> **Thanks to Reviewer LiAP**
>
> Please accept our appreciation for your valuable comments, and in particular for recognizing the strengths of our paper in terms of good motivation, novelty, well-designed methods and metrics, effectiveness, and resistance to standard backdoor defenses.
>
> Please kindly let us know if our response and the new experients have properly addressed your concerns. We are happy to address them before the rebuttal ends.

---

> ### Comment · Reviewer_LiAP · 2022-08-05
> **Thanks for your answer. I have another question.**
>
> Thanks to the authors for your response. The answers addressed my questions pre-rebuttal.
>
> However, I am concerned about one issue raised by another reviewer. In Tables 1 & 2, BA dropped a lot for both targeted and untargeted watermarks. I think the authors picked a training configuration that prioritized ASR over BA. However, if BA is too low, the model will not be picked to use in practice. I wonder if the authors can try other configurations that guarantee the BA drop no more than 2% and check if the watermarked models are still verifiable?

---

> > ### Author Response · Authors · 2022-08-05
> > **Thank you for your feedback and the further insightful question**
> >
> > We greatly appreciate your positive feedback of our rebuttal and the further insightful question. Please kindly find our explanations about the remaining concern below.
> >
> > ---
> > **Q1**: In Tables 1 & 2, BA dropped a lot for both targeted and untargeted watermarks. I think the authors picked a training configuration that prioritized ASR over BA. However, if BA is too low, the model will not be picked to use in practice. I wonder if the authors can try other configurations that guarantee the BA drop no more than 2% and check if the watermarked models are still verifiable?
> >
> > **R1**: Thank you for this insightful question and we do understand your concern. Firstly, we are deeply sorry for the misunderstandings that our response to Reviewer RZea (Table 1-2) may cause you. Specifically, **our Table 2 in the rebuttal may mislead you to think that our UBW-C causes approximately 20% BA drop on the large-scale dataset, compared with our UBW-P**. However, as we mentioned in R5, **the UBW-P was trained on the whole ImageNet while UBW-C was trained on Tiny-ImageNet** (due to the limitation of time and computational resources). Accordingly, comparing their results are meaningless. To verify that our methods (UBW-P and UBW-C) will not significantly reduce the BA of the watermarked models, we provide additional results as follows:
> >
> > Table 1. The performance of our UBW-P on the whole ImageNet dataset.
> > | Attack$\downarrow$, Metric$\rightarrow$ |   BA   |  ASR-A |  ASR-C |  $D_p$  |
> > |:---------------------------------------:|:------:|:------:|:------:|:-------:|
> > |                  No Attack                  | 72.29  | NA  | NA  | NA  |
> > |                  UBW-P                  |    71.36   |    50.00   |    42.56   |    1.8346    |
> >
> >
> > Table 2. The performance of our UBW-C on the Tiny-ImageNet dataset.
> > | Attack$\downarrow$, Metric$\rightarrow$ |   BA   |  ASR-A |  ASR-C |  $D_p$  |
> > |:---------------------------------------:|:------:|:------:|:------:|:-------:|
> > |                  No Attack                  | 54.04  | NA  | NA  | NA  |
> > |                  UBW-C                  | 51.56  | 88.00  | 86.54  | 2.9871  |
> >
> >
> > Besides, as you mentioned, there is a trade-off between the BA and the ASR to some extent, due to the poisoning rate $\gamma$ that users may adopt. As we illustrated in Figure 4 in our main manuscript, the BA decrease while the ASR increase with the increase of $\gamma$. The BA drop is no more than 3% while the ASR is higher than 50% for both UBW-P and UBW-C if we set $\gamma=2\%$ on the CIFAR-10 dataset.
> >
> > Sorry again for the misunderstandings that our rebuttal may cause you and we will add more details and discussions in the appendix of our revision.

---

> > > ### Comment · Reviewer_LiAP · 2022-08-06
> > > **Thanks for your answer**
> > >
> > > Thanks for your answer. It addressed my concern.
> > > I will keep my Accept decision, but the score I will decide later.

---

> > > > ### Author Response · Authors · 2022-08-06
> > > > **Thank You for Your Positive Feedback!**
> > > >
> > > > Thank you again for your valuable time, comments, and suggestions, which greatly help to improve the quality of our paper. Your recognition of our paper also encourages us a lot. Looking forward to your decision on the final score :)

---

### Official Review · Reviewer_UWLP · 2022-07-12

**Rating:** 8
**Confidence:** 4
**Soundness:** 3 good
**Presentation:** 3 good
**Contribution:** 3 good

**Summary:**

This paper proposes a dataset protection approach, Untargeted Backdoor Watermark, via crafting untargeted backdoor attacks for the protected dataset. The authors propose their method in two different settings(i.e., mislabel and clean-label settings). Through extensive experiments,  Untargeted Backdoor Watermark is proved to be effective and robust against various defense approaches in two different settings.

**Questions:**

1. Can you add more descriptions regarding datasets protection and encryption?

**Limitations:**

1. Can u give some practical scenarios to illustrate the importance of UBW.

**Strengths And Weaknesses:**


================= Strength=================
1. The studied problem is interesting.
2. The evaluation is comprehensive.
3. The overall presentation is good.


=================Weakness================

1.   Lack some in-depth discussions on the selection of triggers and provide a practical scenario for data leakage.
2.  It lacks a discussion on the potential negative impact brought by the protected dataset. For example, will the model trained through the protected dataset perform vulnerable against backdoor attacks?

---

> ### Author Response · Authors · 2022-08-02
> **Author Response (Part I)**
>
> We sincerely thank you for your valuable time and comments. We are encouraged by your positive comments on our **research significance**, **extensive experiments**, **method effectiveness**, and **paper writing**. We will alleviate your remaining concerns as follows:
>
>
> ---
> **Q1**: Lack some in-depth discussions on the selection of triggers.
>
> **R1**: Thank you for your insightful comments and we do understand your concerns. We adopted the white-black trigger patterns in the main manuscript simply because it is the most classical one used in existing backdoor-related papers. We have also evaluated the effectiveness of other trigger patterns with diffferent appearances and sizes in Appendix D (Table 1-2). The results show that
>
> - Similar to existing backdoor attacks, both UBW-P and UBW-C can reach promising performance with arbitrary user-specified trigger patterns.
> - The attack success rate increases with the increase of trigger size while its increase has only minor adverse effects on benign accuracy.
>
> Here we quote Table 1-2 for you to refer. Please find more details and discussions in Appendix D.
>
> Table 1. The effectiveness of our UBW with different trigger patterns on the CIFAR-10 dataset.
>
> | Method$\downarrow$ | Pattern$\downarrow$, Metric$\rightarrow$ | BA (\%) | ASR-A (%) | ASR-C (\%) |  $D_p$ |
> |:------------------:|:----------------------------------------:|:-------:|:---------:|:----------:|:------:|
> |        UBW-P       |                Pattern (a)               |  90.59  |   92.30   |    92.51   | 2.2548 |
> |        UBW-P       |                Pattern (b)               |  90.31  |   84.53   |    82.39   | 2.2331 |
> |        UBW-P       |                Pattern \(c\)               |  90.21  |   87.78   |    86.94   | 2.2611 |
> |        UBW-C       |                Pattern (a)               |  86.99  |   89.80   |    87.56   | 1.2641 |
> |        UBW-C       |                Pattern (b)               |  86.25  |   90.90   |    88.91   | 1.1131 |
> |        UBW-C       |                Pattern \(c\)               |  87.78  |   81.23   |    78.55   | 1.0089 |
>
>
> Table 2. The effectiveness of our UBW with different trigger sizes on the CIFAR-10 dataset.
> | Method$\downarrow$ | Trigger Size$\downarrow$, Metric$\rightarrow$ | BA (\%) | ASR-A (%) | ASR-C (\%) |  $D_p$ |
> |:------------------:|:---------------------------------------------:|:-------:|:---------:|:----------:|:------:|
> |        UBW-P       |                       2                       |  90.55  |   82.60   |    82.21   | 2.2370 |
> |        UBW-P       |                       4                       |  90.37  |   83.50   |    83.30   | 2.2321 |
> |        UBW-P       |                       6                       |  90.43  |   86.30   |    86.70   | 2.2546 |
> |        UBW-P       |                       8                       |  90.46  |   86.40   |    86.26   | 2.2688 |
> |        UBW-P       |                       10                      |  90.72  |   86.10   |    85.82   | 2.2761 |
> |        UBW-P       |                       12                      |  90.22  |   88.30   |    87.94   | 2.2545 |
> |        UBW-C       |                       2                       |  87.34  |    4.38   |    15.00   | 0.7065 |
> |        UBW-C       |                       4                       |  87.71  |   70.80   |    64.86   | 1.2924 |
> |        UBW-C       |                       6                       |  87.69  |   75.60   |    70.85   | 1.7892 |
> |        UBW-C       |                       8                       |  88.89  |   75.40   |    69.86   | 1.2904 |
> |        UBW-C       |                       10                      |  88.30  |   77.60   |    73.92   | 1.7534 |
> |        UBW-C       |                       12                      |  89.29  |   98.00   |    97.72   | 1.1049 |
>
> ---
>
> **Q2**: It lacks a discussion on the potential negative impact brought by the protected dataset. For example, will the model trained through the protected dataset is vulnerable to backdoor attacks?
>
> **R2**: Thank you for your insightful comment! As we illustrated in Section 6 (Societal Impacts), we notice that our untargeted backdoor watermark (UBW) is resistant to existing backdoor defenses and could be maliciously used by the backdoor adversaries. However, compared with existing targeted backdoor attacks, our UBW is untargeted and therefore has minor threats. Moreover, although an effective defense is yet to be developed, people can still mitigate or even avoid the threats by only using trusted training resources. Our next step is to explore principled and advanced defenses against UBW-like watermarks.
>
> ---

---

> > ### Author Response · Authors · 2022-08-02
> > **Author Response (Part II)**
> >
> >
> > ---
> > **Q3**: Can you add more descriptions regarding dataset protection and encryption?
> >
> > **R3**: Thank you for this constructive suggestion!
> > - **Dataset protection** has always been an important and wide research area. In this paper, we focus on the protection of released datasets (*e.g.*, open-sourced datasets and commercial datasets). In particular, those datasets are released and can only be used for specific purposes. For example, open-sourced datasets are available to everyone while most of them can only be adopted for academic or educational rather than commercial purposes. Our goal is to detect and prevent unauthorized users of released datasets. This task is challenging since the adversaries can get access to the victim dataset while unauthorized users will only release their trained models without disclosing their training details.
> > - **Encryption** is the most classical protection method, which encrypts the whole or parts of the protected data. Only authorized users who have obtained the secret key can decrypt the encrypted data. However, the encryption can not be exploited to protect released datasets for it will hinder dataset functionalities (*i.e.*, users can not use encrypted dataset for training).
> >
> > We will add more details in our related work in the revision.
> >
> > ---
> > **Q4**: Can you give some practical scenarios to illustrate the importance of UBW?
> >
> > **R4**: Thank you for this constructive suggestion! As we illustrated in the aforementioned R3, our goal is to detect and prevent unauthorized users of released datasets. Specifically, we consider the hardest black-box verification setting where defenders can only get model predictions whereas having no information about its model parameters. This setting is more practical, compared with the white-box one, allowing defenders to perform ownership verification even when they only have access to the model API. Accordingly, **given a suspicous third-party model, we can verify whether it was trained on our protected dataset (without unauthorization) when we can obtain its source files or just the model API**. We will add more details in our related work in the revision.
> >
> > ---

---

> > > ### Comment · Reviewer_UWLP · 2022-08-02
> > > **Post-Rebuttal Review**
> > >
> > > I would like to thank the authors for their detailed feedback. The authors have addressed all my concerns. I read the author's rebuttal and comments from other reviewers. I believe that the proposed methods are novel and elegant. I think it will also have high impact on dataset protection and image watermarking, which are important research areas. Accordingly, I increase my score to 8.

---

> > > > ### Author Response · Authors · 2022-08-03
> > > > **Thank You for Your Positive Feedback!**
> > > >
> > > > Thank you so much for the positive feedback! It encourages us a lot.

---

### Review · Ethics_Reviewer_8Gqd · 2022-07-31

**Recommendation:** None.

**Ethics Review:**

No ethical concerns.

---

### Review · Ethics_Reviewer_Cdcp · 2022-08-18

**Recommendation:** There are no ethical issues in my opi…

**Ethics Review:**

There are no ethical issues in my opinion.

---

### Meta-Review · Area_Chair_DFhZ · 2022-08-28

**Recommendation:** Accept
**Confidence:** Certain

**Metareview:**

This paper proposes a methods to verify unauthorized use of open-sourced dataset. The idea is to inject verifiable backdoor watermarks. The authors first show that existing backdoor watermarks can be exploited by adversaries for attacks. They then proposed novel untargeted backdoor watermarking techniques that are both effective and harmless in poisoned-label (UBW-P) and clean-label (UBW-C) settings. A malicious network that trained using the watermarked dataset may predict randomly for watermarked test data and clearly for clean test data, so it is possible to verify using the difference between the two predictions, for watermarked and clean test data. The authors agree that the proposed untargeted watermarks are useful and the problem being studied is interesting. The authors are suggested to address remaining concerns of the reviewers, such as whether the random classification is better than previous guided misclassification for verifying malicious users.

**Award:**

No

---

### Decision · Program_Chairs · 2022-09-14

Accept